# A comparison of transporter gene expression in three species of *Peronospora* plant pathogens during host infection

Eric T. Johnson[1]*, Rebecca Lyon[1], David Zaitlin[2], Abdul Burhan Khan[3], Mohammad Aman Jairajpuri[3]

**1** United States Department of Agriculture, Agricultural Research Service, National Center for Agricultural Utilization Research, Crop Bioprotection Unit, Peoria, Illinois, United States of America, **2** Kentucky Tobacco Research & Development Center, University of Kentucky, Lexington, Kentucky, United States of America, **3** Department of Biosciences, Jamia Millia Islamia University, New Delhi, India

* eric.johnson2@usda.gov

**Data Availability Statement:** All raw sequencing data are deposited in the National Center for Biotechnology Information under accession code PRJNA852505 (https://www.ncbi.nlm.nih.gov/

## Abstract

Protein transporters move essential metabolites across membranes in all living organisms. Downy mildew causing plant pathogens are biotrophic oomycetes that transport essential nutrients from their hosts to grow. Little is known about the functions and gene expression levels of membrane transporters produced by downy mildew causing pathogens during infection of their hosts. Approximately 170–190 nonredundant transporter genes were identified in the genomes of *Peronospora belbahrii*, *Peronospora effusa*, and *Peronospora tabacina*, which are specialized pathogens of basil, spinach, and tobacco, respectively. The largest groups of transporter genes in each species belonged to the major facilitator superfamily, mitochondrial carriers (MC), and the drug/metabolite transporter group. Gene expression of putative *Peronospora* transporters was measured using RNA sequencing data at two time points following inoculation onto leaves of their hosts. There were 16 transporter genes, seven of which were MCs, expressed in each *Peronospora* species that were among the top 45 most highly expressed transporter genes 5–7 days after inoculation. Gene transcripts encoding the ADP/ATP translocase and the mitochondrial phosphate carrier protein were the most abundant mRNAs detected in each *Peronospora* species. This study found a number of *Peronospora* genes that are likely critical for pathogenesis and which might serve as future targets for control of these devastating plant pathogens.

## Introduction

The genus *Peronospora* contains many plant pathogens that cause damage to crops and ornamentals. There were approximately 400 described *Peronospora* species in the late 20th century [1], but one study estimated that there could be 3,000–5,000 species of *Peronospora* worldwide [2]. All species of *Peronospora* are obligate biotrophs, which means they are completely dependent on the living tissues of their host plant for nutrition. Several of the most economically relevant *Peronospora* species have been studied in detail. For example, *Peronospora effusa* causes

**Funding:** Funding for this work was provided to EJ and RL using United States Department of Agriculture, Agricultural Research Service in-house project 5010-22410-017-00-D. The funders had no role in study design, data collection and analysis, decision to publish, or preparation of the manuscript.

**Competing interests:** The authors have declared that no competing interests exist.

a serious disease of cultivated spinach. The pathogen possibly grows systemically in the plant and can be transmitted via the seed [3]. Synthetic fungicides, in combination with host resistance genes, can manage *P. effusa* [4–6], but this is not an option for organic spinach production. In addition, 17 races of *P. effusa* have been identified as of 2018. [7]. The genomes of *P. effusa* races 13 and 14 have been sequenced and described [7]. Another problem species is *Peronospora belbahrii*, which grows on sweet basil. The disease was first identified in Switzerland in 2001 [8] but can be found worldwide now, most likely due to the dissemination of infected seeds [9]. *P. belbahrii* causes yellowing of infected leaves, and then sporangia grow out of stomata on the undersides of leaves 7–10 days after inoculation [10,11]. Synthetic fungicides are the primary means to control downy mildew disease in basil [12–15]. Control of downy mildew disease in basil was achieved in several cases through the action of two or more genes, but the mechanism(s) of host resistance is undetermined [16–18]. The genome of a German isolate of *P. belbahrii* was sequenced and described [19]. Another downy mildew disease in many parts of the world is tobacco blue mold disease, caused by *Peronospora tabacina*, that can result in substantial financial losses [2]. *P. tabacina* can induce lesions on leaves, and there is additional evidence that the pathogen can grow systemically [20,21]. This tobacco pathogen is controlled by synthetic fungicides at present, but resistance in the pathogen has been documented [22,23]. Genomes from two German isolates of *P. tabacina* were sequenced several years ago [24].

Analyses of the five *Peronospora* genomes indicated that there were approximately 8,000 gene models in *P. tabacina* and *P. effusa* whereas there were slightly more than 9,000 gene models in *P. belbahrii* [7]. It was determined that 4,148 gene models were unique to one or the other of the two sequenced *P. tabacina* isolates, which differed in their sensitivity to the fungicide metalaxyl-M [7,24]. In addition, 3,095 genes were unique to *P. tabacina* compared to other oomycetes, although *P. belbahrii* was not included in this analysis [7]. The mitochondrial sequences of the two *P. tabacina* isolates had sequence variations, including seven single nucleotide polymorphisms, three indels, and a difference in the copy number of a repeated sequence [24]. In contrast, the genome sequences of the two *P. effusa* isolates were more similar to each other, with only 1,415 genes unique to one isolate or the other, and the *P. effusa* mitochondrial genome sequences were identical [7]. Analysis of *P. effusa* gene models indicated that 1,807 were unique compared to other oomycetes, although *P. belbahrii* was not included in the comparison [7]. Although extensive comparison of the *P. belbahrii* genome with other oomycete genomes was not done, hierarchical clustering analysis of metabolic networks from 11 oomycetes determined that the network of *P. belbahrii* clustered with oomycete networks from obligate biotrophs (including *Albugo laibachii*, *Plasmopara halstedii*, and *Hyaloperonospora arabidopsidis*) whereas the hemibiotrophs, primarily *Phytophthora* species, grouped together in a different clade [19]. In addition, the hierarchical clustering analysis indicated that the metabolic networks in obligate biotrophs are generally smaller than the metabolic networks of hemibiotrophs [19]. A similar pattern of clustering of the metabolic networks of obligate plant pathogens was found when principal component analysis of metabolic networks was performed with the genomes of 42 oomycetes, six of which were obligate biotrophs (*A. laibachii*, *H. arabidopsidis*, *P. belbahrii*, *P. effusa*, *P. halstedii*, and *Plasmopara viticola*) [25]. The losses of metabolic genes in the obligate plant pathogens were typically in the same sets of metabolic enzymes despite the fact they were classified into four different genera [25].

Obligate biotrophs cannot be cultured on artificial medium in the laboratory. Analysis of the first genome sequence of a downy mildew causing pathogen (*Hyaloperonospora arabidopsidis*) indicated that it lacked the ability to reduce nitrate and sulfite [26]. This same scenario was also found in one isolate of *P. tabacina*, and one isolate of *Plasmopara halstedii*, a downy mildew causing pathogen of sunflower [7]. Interestingly, one isolate of *P. tabacina* only lacked

nitrate reductase, and not sulfite reductase; this was also true in two isolates of *P. effusa* [7]. These gene deficiencies strongly suggest that the host plant is a sufficient source of nitrogen and sulfate [27]. Additional analyses indicated that the genomes of *P. effusa* and *P. tabacina* have reduced numbers of genes that encode proteins for carbohydrate, calcium, flagella-motor, phytopathology, and transporter activities compared to the genomes of *Phytophthora* species [7]. The reduction in the numbers of genes encoding different transporters suggests that *Peronospora* species are limited in the host substrates that can be transported into hyphae or sporangia, but the expression of genes encoding transporters, as well as the possible substrates of the transporters, has been only documented for a select a group of transporters. The aim of this study was to identify all the transporter genes in three *Peronospora* species and determine which of these genes are well expressed during infection of their host. This study of *Peronospora* transporter gene expression could be valuable in formulating crop protection products in the future, and contributes to a better understanding of the key genes involved in oomycete nutrient acquisition.

## Materials and methods

### Plant materials and propagation of the *P. tabacina* pathogen

The pathogen *P. tabacina* was maintained on tobacco variety KY14 as previously described [28]. Briefly, plants were routinely inoculated with *P. tabacina* by spraying them with a sporangia suspension of ~100,000/ml. The inoculated plants were placed in large plastic totes, with the lids sealed, which were kept in an isolation chamber maintained at 21˚C. The plants were removed from the totes and placed on shelves in the isolation chamber the next day and grown for 6 days at 21˚C with a 12-hour photoperiod. Leaves showing signs of infection were harvested and sealed in a large baggie or a plastic box with moist paper towels and placed in the dark in the same chamber for 16 hours to allow the pathogen to sporulate. The sporangia were brushed off the undersides of the leaves with a camelhair brush (~1 inch width) into a flat container (Pyrex baking dish) containing highly purified deionized water. The sporangia were used directly for re-inoculation or were collected by filtration on a 5-micron membrane (Millipore-Sigma, Burlington, MA) and washed several times prior to resuspension in water and counting with a hemacytometer. The washed sporangia were used in experiments where it was important to know the concentration.

### Inoculation of tobacco leaves with *P. tabacina*

Five nonflowering KY14 tobacco plants, approximately four months old, were inoculated in at least 30 places per plant (four-five leaves per plant) by infiltration of 50–100 μl of *P. tabacina* sporangia (100,000 per ml) using a 1 ml syringe. The plants were sampled at 2 and 5 days post-inoculation (DPI); only samples from diseased plants were processed. Eight tissue samples per plant were excised using a 2-cm diameter cork borer from each inoculation site. The circular disks were immediately frozen in liquid nitrogen and stored at -80˚C.

### RNA extraction and transcriptome sequencing of tobacco leaves

RNA was extracted from frozen leaves as previously described [29]. Briefly, leaf tissue was powdered with a mortar and pestle using liquid nitrogen. The powdered tissue was placed in a 1.5 ml centrifuge tube that contained one ml of TRIzol (Thermo Fisher Scientific, Waltham, MA). RNA was purified from the solution using the TRIzol Plus RNA Purification Kit (Thermo Fisher Scientific). The RNA was incubated with DNAse (Qiagen, Hilden, Germany) to remove any trace genomic DNA and purified using the GeneJET RNA Cleanup Kit

(Thermo Fisher Scientific). The RNA samples (2 DPI, three samples, 5 DPI, four samples) were shipped to Azenta Life Sciences (South Plainfield, NJ) for transcriptome sequencing. Quality control analysis of all the samples was performed by Azenta Life Sciences using TapeStation (Agilent Technologies, Palo Alto, CA), which found that the RIN values were >4.0 and the DV200 scores (percentage of RNA fragments >200 base pairs in length) were >70% (S1 Table), which indicated that the quality and integrity of the RNA samples were sufficient for transcriptome sequencing [30].

## Mapping of tobacco transcriptomes to the *P. tabacina* genome

The raw reads were trimmed and mapped using the "map reads to reference" tool in QIAGEN CLC Genomics Workbench version 22.0.1. Alignment of the reads to the *P. tabacina* genome (GenBank: GCA_002099245 [24]) was performed using default settings. In other analyses, a single gene served as the reference for read mapping of transcriptomes using the same tool with default settings.

## Identification of putative transporter genes

Sequences of putative transporter proteins from the *P. effusa* genome, which were organized into 19 different groups [31], were downloaded from the InterPro website [32]. This set of transporter proteins is not comprehensive because it depends on high quality genome annotation. The CD-HIT suite [33] was utilized to identify highly homologous proteins in this set; a protein was removed from the set if it had 90% identity to another protein, and the longest protein (of a homologous pair or group) was retained. The gene sequences of the putative *P. effusa* transporters were used to find the homologous genes in *P. belbahrii* (using BLAST searches of the genome sequence, GCA_902712285.1 [19], on CLC Genomics Workbench) and *P. tabacina* (using BLAST searches on the NCBI website). The protein sequences were screened for a transporter motif using the Conserved Domain Database from NCBI and hmmscan from EMBL-EBI; only proteins that had a transporter motif score of $10^{-5}$ were retained in the set. All the motif scores are available in S2 Table.

## Gene expression analysis

A gene expression value (TPM, transcripts per million) was calculated for each putative transporter gene and seven housekeeping genes (S1 File) in each transcriptome sample. Transcriptome read files from susceptible KY14 burley tobacco infected with *P. tabacina* at 2 and 5 DPI (described above), susceptible spinach cultivar 'Viroflay' infected with *P. effusa* at 2 and 7 DPI [34], and susceptible basil infected with *P. belbahrii* at 3 and 6 DPI [29] were included in the expression analysis. The data from each transcriptome were normalized using the trimmed mean of M-values normalization method [35].

## Phylogenetic and protein analysis

Protein sequences were aligned and phylogenetic trees were constructed using MEGA version 10 [36]. Protein alignments and a percent identity matrix of transporters was calculated using Clustal Omega [37]. The number of transmembrane helices in proteins was estimated using several software programs including HMMTOP [38], deep TMHMM [39], and SPLIT 4.0 [40]. The TM-align algorithm [41] was utilized to compare two modeled protein structures. The EMBOSS needle alignment algorithm [42], which is available on the EMBL- EBI server [43], was used for comparing two protein sequences.

### *Ab Initio* modelling of protein sequences

I-TASSER (Iterative Threading Assembly Refinement; https://zhanglab.ccmb.med.umich.edu/I-TASSER/) was used to predict protein structure and function. I-TASSER creates a 3D model of a protein using an *ab initio* modelling method [44]. The C-score is a confidence score used by I-TASSER to estimate the superiority of projected models; it is normally in the range of -5 to 2, with a higher number indicating a model with high confidence [45,46].

## Results and discussion

### Identification of putative transporter genes in three *Peronospora* species

This study focused on the identification of transporter genes primarily involved in providing nutrition for *Peronospora* pathogens and excluded the ABC transporters and ion channels. Nearly 200 putative transporter genes were identified in the pathogen *P. effusa* (Tables 1 and S2). Fewer transporter genes were identified in *P. belbahrii* and *P. tabacina* for similar reasons. In a few cases, stop codons could be found in the *P. belbahrii* or *P. tabacina* genes that were homologous to those from *P. effusa* (S2 Table). In other cases, no homologous genes could be found in *P. belbahrii* or *P. tabacina*. Lastly, no conserved transporter motifs were identified in the protein encoded by the gene, or the motif identified lacked statistical support.

The Major Facilitator Superfamily (MFS) of transporters had the highest number of gene members in all three *Peronospora* species compared to the other transporter families (Table 1). The MFS family is also the largest transporter family in other plant pathogens such as *Phytophthora infestans*, *Pythium ultimum*, and *Magnaporthe oryzae* [31]. It should be noted that the Folate-Biopterin, Glycoside-Pentoside-Hexuronide:Cation Symporter, and Proton-dependent Oligopeptide Transporter families are specialized MFS transporters. There were also a number of gene members from the Mitochondrial Carrier and Drug/Metabolite Transporter families identified in the genomes of the three *Peronospora* species.

Table 1. Number of putative transporters in each of the three *Peronospora* species included in this study.

| Transporter | *P. effuse* | *P. belbahrii* | *P. tabacina* |
|---|---|---|---|
| Amino Acid-Polyamine Organocation (APC) | 9 | 9 | 9 |
| Amino Acid/Auxin Permease (AAAP) | 15 | 12 | 15 |
| Ammonia Transporter (AMT) | 1 | 1 | 1 |
| Choline Transporter-Like (CTL) | 6 | 6 | 6 |
| Dicarboxylate/Amino Acid Cation Symporter (DAACS) | 5 | 4 | 4 |
| Drug/Metabolite Transporter (DMT) | 36 | 30 | 30 |
| Equilibrative Nucleoside Transporter (ENT) | 3 | 3 | 3 |
| Folate-Biopterin Transporter (FBT) | 15 | 13 | 13 |
| Glycoside-Pentoside-Hexuronide:Cation Symporter (GPH) | 1 | 2 | 1 |
| Major Facilitator Superfamily (MFS) | 45 | 40 | 40 |
| Mitochondrial Carrier (MC) | 35 | 33 | 33 |
| Multidrug/Oligosaccharidyl-lipid/Polysaccharide Flippase (MOP) | 5 | 4 | 4 |
| Phosphate:Na $^+$ Symporter (PNaS) | 1 | 1 | 1 |
| Proton-dependent Oligopeptide Transporter (POT) | 2 | 2 | 2 |
| Solute:Sodium Symporter (SSS) | 3 | 3 | 2 |
| Sulfate Permease (SulP) | 8 | 9 | 7 |
| SWEET-MtN3 (SWEET) | 2 | 2 | 2 |
| Total | 192 | 174 | 173 |

## Differential expression of *Peronospora* transporter genes during interactions between oomycete pathogens and their susceptible hosts

Expression of all putative transporter genes in *P. effusa* was analyzed in spinach leaves during susceptible host infection (S3 Table). The TPM levels of the transporter genes of *P. effusa* were quite low at 2 DPI; the proportion of total RNA mapped to the *P. effusa* genome was <1% in these leaf samples [34]. Only one transporter gene, RMX67221, had enhanced expression at 2 DPI, although it was ~50% the level of the tubulin B housekeeping gene (Table 2). Many more transporter genes were expressed by *P. effusa* at 7 DPI (S3 Table); this finding is likely due to

**Table 2. Transporter protein genes in the three *Peronospora* species that were highly expressed (> 10,000 transcripts per million, TPM) in one and (or) two of the sampled leaf tissues.**

| *P. effuse* | 2 DPI[a] TPM ± SD (N = 6) | 7 DPI TPM ± SD (N = 4) |
|---|---|---|
| Actin | 451,740 ± 29,072 | 117,290 ± 9,241 |
| beta tubulin | 40,226 ± 4,212 | 18,917 ± 2,220 |
| pyruvate DHα[b] | 412 ± 364 | 13,784 ± 888 |
| RMX62118[c] (AAAP)[d] | 319 ± 145 | 14,714 ± 5,969 |
| RMX65655 (MC) | 106 ± 97 | 13,237 ± 1,463 |
| RMX67221 (MC) | 23,647 ± 2,313 | 51,876 ± 5,203 |
| RMX62154 (MFS) | 124 ± 198 | 11,245 ± 1,553 |
| RQM16367 (MFS) | 230 ± 280 | 19,601 ± 2,683 |
| *P. belbahrii* | 3 DPI TPM ± SD (N = 5) | 6 DPI TPM ± SD (N = 4) |
| Actin | 205,745 ± 24,977 | 123,359 ± 26,948 |
| beta tubulin | 19,406 ± 2,462 | 14,625 ± 2,452 |
| pyruvate DHα | 11,229 ± 2,730 | 11,997± 3,626 |
| RMX63658 (AAAP) | 1,993 ± 520 | 17,233 ± 2,666 |
| RMX63775 (MC) | 3,336 ± 599 | 20,560 ± 6,600 |
| RMX65655 (MC) | 14,948 ± 3,187 | 15,710 ± 3,554 |
| RMX67221 (MC) | 163,026 ± 28,832 | 156,778 ± 24,392 |
| RMX67668 (MC) | 12,394 ± 1,403 | 6,914 ± 1,791 |
| RMX68742 (CTL) | 5,767 ± 1,114 | 23,332 ± 930 |
| RMX67547 (MFS) | 7,850 ± 696 | 14,117 ± 1,589 |
| RQM16367 (MFS) | 9,922 ± 2,283 | 7,401 ± 1,569 |
| *P. tabacina* | 2 DPI TPM ± SD (N = 3) | 5 DPI TPM ± SD (N = 4) |
| actin | 143,432 ± 11,923 | 159,111 ± 18,002 |
| beta tubulin | 23,884 ± 2,128 | 20,665 ± 2,327 |
| pyruvate DHα | 17,639 ± 1,350 | 15,530 ± 2,509 |
| RMX62154 (MFS) | 5,399 ± 1,378 | 10,841 ± 922 |
| RQM16367 (MFS) | 12,092 ± 1,434 | 18,489 ± 2,420 |
| RMX65655 (MC) | 26,021 ± 3,976 | 24,588 ± 2,434 |
| RMX67221 (MC) | 98,033 ± 11,117 | 69,417 ± 7,722 |
| RMX67668 (MC) | 10,937 ± 867 | 13,641 ± 3,805 |
| RMX68746 (DMT) | 19,286 ± 3,351 | 21,093 ± 4,239 |
| RQM14033 (DAACS) | 10,342 ± 1,448 | 16,771 ± 1,325 |

[a]DPI is days post inoculation.

[b]DHα is dehydrogenase α subunit.

[c]Numbers preceded by RMX and RQM are *P. effusa* gene accession numbers given by the European Nucleotide Archive (ENA). For comparative purposes in this study, homologous genes of *P. belbahrii* and *P. tabacina* were assigned the ENA gene accession number of *P. effusa*.

[d]Abbreviations for each transporter classification are defined in Table 1.

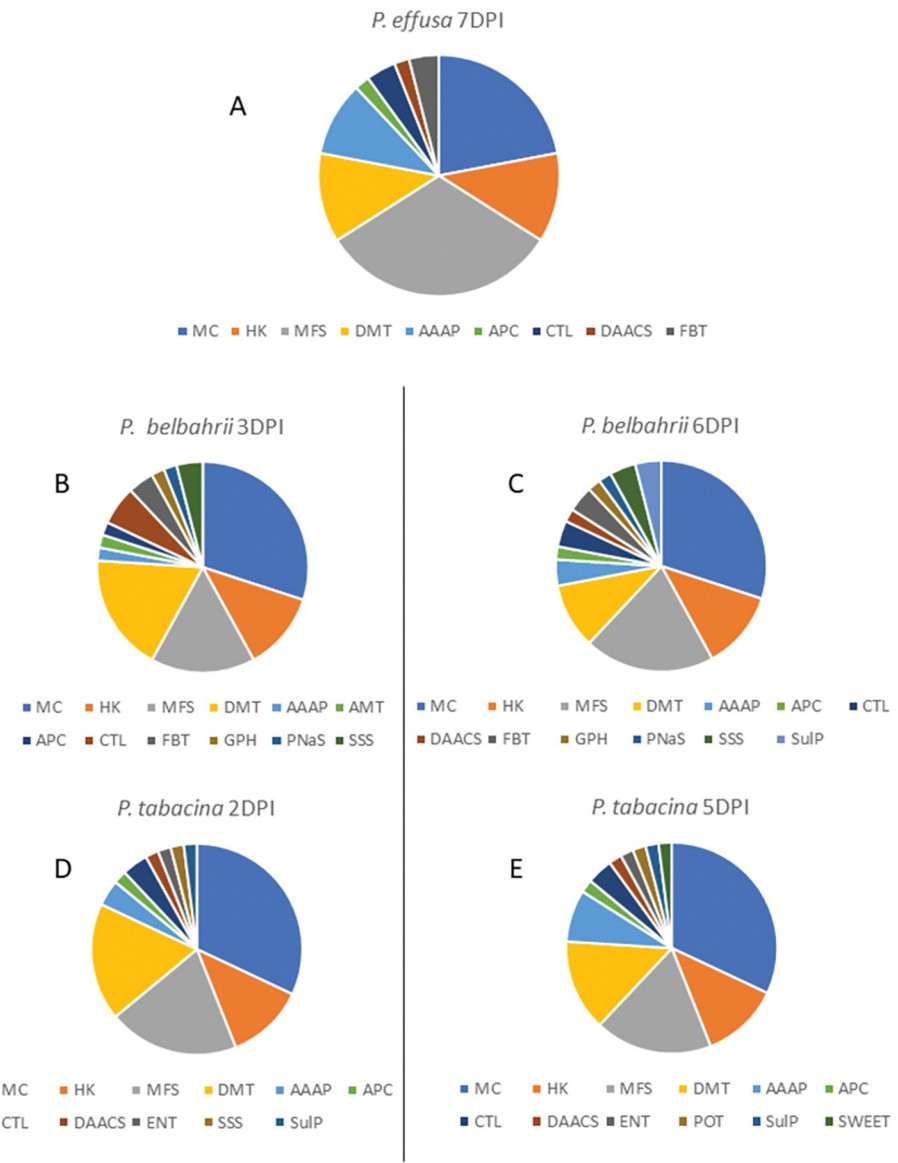

**Fig 1. Types of transporter genes expressed in three *Peronospora* pathogens during host infection.** The panels show the distribution of the top 50 most highly expressed genes into various transporter classes in each pathosystem at the listed time point; refer to Table 1 for the meaning of the abbreviations. HK stands for housekeeping gene. A, *P. effusa* 7 DPI; B, *P. belbahrii* 3 DPI; C, *P. belbahrii* 6 DPI; D, *P. tabacina* 2 DPI; E, *P. tabacina* 5 DPI. DPI, days post inoculation.

more growth of the pathogen in the plants between 2 DPI and 7 DPI because 25 to 48% of the total sequence reads were mapped to the *P. effusa* genome in these RNA samples [34]. Fifty percent of the top 50 most highly expressed genes had TPM values >5,000 and included the housekeeping genes (S3 Table). The class of transporters that was most frequently found in the top 50 most highly expressed genes encoded MFS transporters, followed by the Mitochondrial Carrier (MC) transporters (Fig 1A). In addition, the TPM levels of two MFS, two MC and one Amino Acid/Auxin Permease (AAAP) gene were close to the same TPM levels of the housekeeping genes beta tubulin and pyruvate dehydrogenase alpha subunit in *P. effusa* at 7 DPI (Table 2).

The expression rates of transporter genes analyzed for *P. effusa* were also measured in RNA samples extracted from basil leaves infected by *P. belbahrii* [29], as detailed in S4 Table; approximately 17% and 32% of the total reads mapped to the *P. belbahrii* genome in the 3 and 6 DPI replicates, respectively [29]. Slightly less than 50% of the top 50 most highly expressed genes had TPM values >5,000 at 3 DPI whereas 60% of the top 50 most highly expressed genes were >5,000 TPM at 6 DPI (S4 Table). All of the *P. belbahrii* housekeeping genes had TPM values of least 4,000 in all of the 3 and 6 DPI samples (S4 Table). The expression levels of *P. belbahrii* actin and RMX67221 exceeded 30,000 TPM at both 3 and 6 DPI (S4 Table). The transporter class that was most frequently found in the top 50 most highly expressed *P. belbahrii* gene transcripts analyzed were MC transporters (Fig 1B and 1C). Some of the most highly expressed *P. belbahrii* gene transcripts are listed in Table 2. Three different *P. belbahrii* MC transporter genes were highly expressed at 3 DPI, especially expression levels of RMX67221. At 6 DPI, the RMX67221 TPM levels were still very high, at approximately the same level as actin; the mRNA expression levels of two other MC genes were close to those of the housekeeping genes beta tubulin and pyruvate dehydrogenase alpha at this time point. In addition, an AAAP, MFS, and a Choline Transporter-Like (CTL) transporter gene had substantial mRNA levels in *P. belbahrii* at 6 DPI.

Lastly, the transporter genes were analyzed from RNA isolated from tobacco leaves infected by *P. tabacina* (S5 Table). The mean mapping rates of the total reads to the *P. tabacina* genome were 20% ± 6 (SD) for the 2 DPI samples (N = 3), and 50% ± 5 for the 5 DPI samples (N = 4). The expression levels of actin and RMX67221 exceeded 30,000 TPM in *P. tabacina* at both 2 and 5 DPI (S5 Table). Approximately 34% and 52% of the top 50 most highly expressed gene transcripts in *P. tabacina* had TPM values >5,000 TPM at 2 and 5 DPI, respectively (S5 Table). All of the *P. tabacina* housekeeping genes had TPM values of at least 3,000 in all the samples (S5 Table). As we found for *P. belbahrii*, the MC transporter genes of *P. tabacina* were the predominant transporter class among the top 50 most highly expressed genes analyzed in *P. tabacina* (Fig 1D and 1E). Some of the most well-expressed (where the expression level was 10,000 TPM or greater at 2 or 5 DPI) *P. tabacina* genes are given in Table 2. Another MC transporter gene, RMX65655, was also well expressed at both 2 and 5 DPI. A variety of other *P. tabacina* transporter genes were transcribed at TPM levels of 10,000 or higher at 5 DPI.

Comparisons among the top 45 most highly expressed transporter genes (not including housekeeping genes) were made between the three *Peronospora* species at the later dates following inoculation (5–7 DPI, Fig 2A). Sixteen transporter genes were well-expressed in each organism at these later dates following inoculation timepoint. Of the 16 transporter genes in common, seven were MCs, as shown in Fig 2B. In addition, there were four MFS transporter genes that were well-expressed in all three *Peronospora* pathogens (Fig 2C). In each species, 11–18 transporter genes were expressed exclusively among the top 45 most highly expressed genes (Fig 2A) which may reflect the diversity of metabolites available to the pathogen in each plant host or may be due to variations in metabolism in each pathogen. A comparison of 91 KEGG [47] metabolic pathways from 42 oomycete proteomes, including *P. effusa* and *P. belbahrii*, was recently published [25]. The "pan-pathways" were defined as those pathways where each reaction was identified in at least one species [25]. Pathway coverage was then calculated by dividing the number of reactions of each species by the total number of reactions in the pan-pathway [25]. The pathway coverage of *P. effusa* and *P. belbahrii* was similar for the majority of the 91 metabolic pathways, but there were notable differences; for example, isoquinoline alkaloid biosynthesis pathway coverage was 14% for *P. belbahrii* and 43% for *P. effusa*; lipopolysaccharide biosynthesis pathway coverage was complete (100%) for *P. belbahrii* but only 10% for *P. effuse* [25]. These differences in pathogen metabolism may require unique

## A. Top 45 most highly expressed transporters

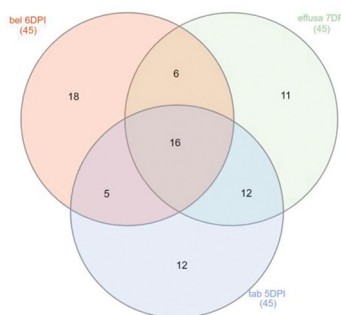

## B. MC transporters among top 45 most highly expressed transporters

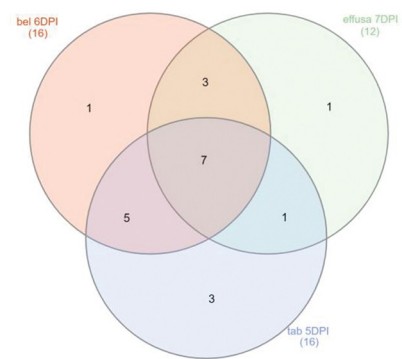

## C. MFS transporters among top 45 most highly expressed transporters

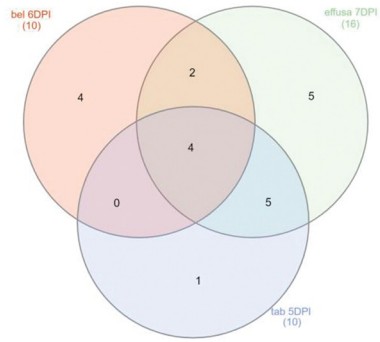

**Fig 2. Venn diagrams of various transporter genes in each *Peronospora* pathosystem at the latest analyzed time point after inoculation.** A, the top 45 most highly expressed transporter genes; B, the number of MC transporters among the top 45 most highly expressed transporter genes; C, the number of MFS transporters among the top 45 most highly expressed transporter genes. DPI, days post inoculation. bel, *P. belbahrii*; effusa, *P. effusa*; tab, *P. tabacina*.

transporters for metabolite import or export. It is also interesting to note that a handful of transporter genes in *Plasmopara viticola*, a downy mildew causing pathogen of grapevine, were under positive selection [48]. Much more research is necessary to identify essential metabolites imported by these three *Peronospora* pathogens from their hosts.

## Possible functions for the highly expressed *Peronospora* transporter genes

The putative functions of the 16 transporter genes that were well-expressed in all three *Peronospora* organisms were identified from the NCBI annotation (Table 3). In the next few paragraphs, the putative functions of the 16 transporter genes will be described and examples from other organisms discussed.

The RMX67221 gene encodes a putative ADP/ATP translocase that is necessary for energy production in the mitochondria of all organisms and is one of the most abundant proteins in the inner mitochondrial membrane [49]. The protein encoded by RMX67221 in all three *Peronospora* species contains the consensus motif RRRMMM, which is typical of all ADP/ATP translocases [50]. The RMX65655 gene encodes a putative mitochondrial phosphate carrier protein, and like ADP/ATP translocases, it is essential for energy production from the mitochondria [51]. The protein encoded by RMX65655 is quite similar to the phosphate carrier in *Arabidopsis thaliana* (BLASTP E value was 2e-97). The results of a previous study indicated that both ADP/ATP translocase and phosphate carrier protein levels were the most abundant proteins in the inner mitochondrial membrane in *A. thaliana* [52].

RMX67457 codes for a putative mitochondrial GTP/GDP carrier protein. The yeast gene orthologous to RMX67457, *GGC1*, is important for maintenance of mitochondrial DNA synthesis in yeast cells [53]. RMX67668 encodes a putative citrate/oxoglutarate carrier protein; the yeast orthologous protein transports citrate and oxoglutarate, and was also shown to transport oxaloacetate, succinate and fumarate [54], suggesting that the best substrate for the *Peronospora* RMX67668 proteins should be tested using *in vitro* techniques. The top BLASTP hit for RMX63170 indicated that this gene encodes a putative ADP/ATP carrier protein, otherwise known as an ADP/ATP translocase. However, lower scoring BLASTP hits suggested that RMX63170 encodes for a putative peroxisomal adenine nucleotide transporter. Phylogenetic

**Table 3. Putative functions of genes commonly expressed in the three *Peronospora* species.**

| MC | Putative function based on BLASTP analysis at the NCBI and Expect value[a] |
|---|---|
| RMX67221 | ADP/ATP translocase [*Phytophthora infestans*] 3e-179 |
| RMX65655 | Mitochondrial phosphate carrier protein 3 [*Phytophthora cactorum*] 0.0 |
| RMX67457 | Mitochondrial GTP-GDP carrier protein 1 [*Plasmopara halstedii*] 0.0 |
| RMX67668 | Citrate/oxoglutarate carrier protein [*Phytophthora ramorum*] 1e-179 |
| RMX63170 | Peroxisomal adenine nucleotide carrier 1 [*Phytophthora ramorum*] 2e-157 |
| RMX66310 | Solute carrier family 25 member 40 [*Phytophthora ramorum*] 1e-166 |
| RMX69262 | Mitochondrial 2-oxoglutarate/malate carrier protein [*Phytophthora infestans*] 0.0 |
| MFS | |
| RMX65758 | Acetyl-coenzyme A transporter [*Phytophthora megakarya*] 0.0 |
| RMX65273 | Glucose transporter [*Phytophthora infestans*] 0.0 |
| RQM16367 | Glucose transporter [*Phytophthora cinnamomi*] 0.0 |
| RMX67545 | Solute carrier family 2, facilitated glucose transporter member 3 [*Phytophthora ramorum*] 0.0 |
| Others | |
| RMX64359 (CTL) | Hedgehog/Intein (Hint) domain [*Phytophthora cactorum*] 0.0 |
| RMX68742 (CTL) | Protein PNS1 [*Phytophthora ramorum*] 0.0 |
| RMX67305 (DMT) | NIPA-like protein 2 [*Phytophthora ramorum*] 0.0 |
| RQM14033 (DAACS) | C4-dicarboxylate transport protein [*Phytophthora ramorum*] 0.0 |
| RMX68746 (DMT) | UAA transporter family [*Phytophthora infestans*] 0.0 |

[a]The most specific annotation possible was chosen.

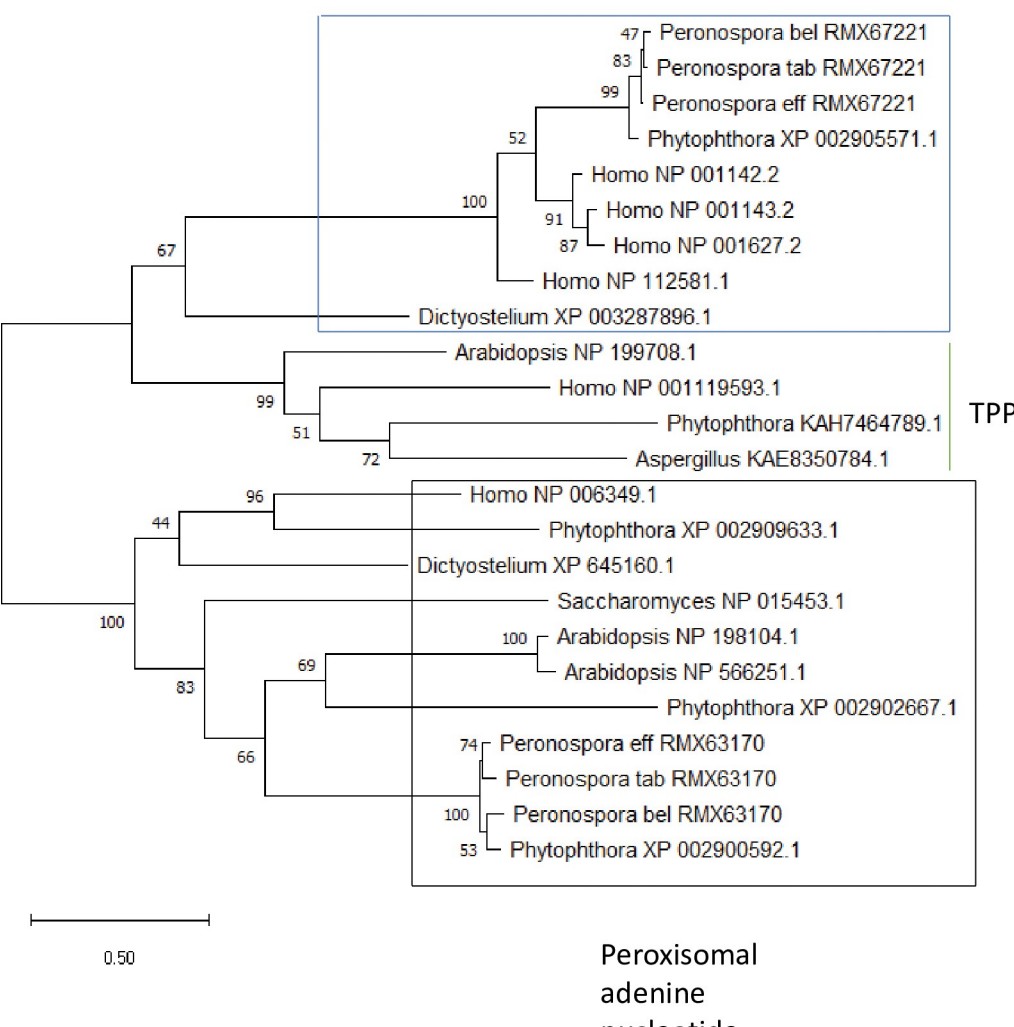

**Fig 3. Phylogenetic analysis of various adenine nucleotide transporter proteins.** The maximum likelihood tree was constructed using the LG model. The percentages of replicate trees in which the associated proteins clustered together in the bootstrap test (1,000 replicates) are shown next to the nodes. The proteins in the blue box are putative ADP/ATP translocases; the proteins in the black box are putative peroxisomal nucleotide transporters; the proteins directly left of the green line are the outgroup, which are putative thiamine pyrophosphate (TPP) transporters. The genus of the organism from which the protein originated and the Genbank number is given at the tip of each branch; a '_' follows each XP or NP. The scale bar indicates the number of protein substitutions per site.

analysis showed that the predicted RMX63170 proteins from the three *Peronospora* species cluster in a clade with other known peroxisomal adenine nucleotide transporters, not in the clade containing the ADP/ATP carrier proteins (Fig 3). The peroxisomal adenine nucleotide transporter moves ATP from the cytosol into the peroxisome in exchange for AMP; the ATP is important for the activation of fatty acids for β-oxidation that occurs in the peroxisome [55].

The RMX66310 gene encodes a putative solute carrier family 25 member 40; the substrates for the orthologous proteins from human and fruit fly have yet to be identified, but the genes are expressed in nerve tissue mitochondria in both of these organisms [56]. RMX69262 encodes a putative 2-oxoglutarate/malate carrier protein, also known as the mitochondrial dicarboxylate-tricarboxylate carrier (DTC) protein [57]. The RMX69262 protein from *P. effusa*

shares 38% identity with *A. thaliana* DTC (At5g19760, BLASTP E value = 2e-67), which specifically transports protonated citrate and unprotonated malate [57]; DTCs can also transport oxaloacetate, oxoglutarate, isocitrate cis-aconitate, and trans-aconitate [57].

RMX65758 encodes a putative acetyl-coenzyme A transporter protein that is localized to the endoplasmic reticulum (ER) in mammalian cells [58]. The acetyl-coenzyme A is used to acetylate membrane proteins in the ER, and defects in the acetyl-coenzyme A transporter have been linked to neurodegenerative disease in humans [59]. Surprisingly, the *P. effusa* acetyl-coenzyme A transporter has 47% amino acid identity with the human protein, known as AT-1. Carbohydrate transporters are part of the MFS superfamily. Three of the four MFS genes listed in Table 3 were predicted to be carbohydrate (specifically glucose) transporters. One of the most highly expressed MFS transporter genes in *P. belbahrii* was RMX67547 (Table 2), which was also among the top 50 most highly expressed transporter genes in *P. effusa* at 7 DPI; however, the RMX67547 gene was not expressed in *P. tabacina* at either 2 or 5 DPI.

The protein encoded by RMX64359 contains a putative Hedgehog/Intein (Hint) domain, but other high scoring annotations included "Calponin homology domain-containing proteinoline transporter", metal transporter CNNM4 (magnesium transporter, also includes CNNM2), and Pns1p (the only choline transporter-like protein (CTL) in *Saccharomyces cerevisiae* [60]). In addition, the RMX68742 gene potentially codes for the Pns1p protein. The Hint domain in the hedgehog protein and the calponin homology domain do not exhibit properties of a membrane transporter [61,62]. We performed a phylogenetic analysis of a sequence alignment that included several putative magnesium transporters and many putative CTLs (Fig 4). All of the *Peronospora* proteins encoded by RMX64359- and RMX68742-type genes were more related to the clade containing Pns1p than they were to the magnesium transporters or the clade containing Ctl1 from the fission yeast *Schizosaccharomyces pombe*, which is required for autophagosome formation [63]. However, one study found that *S. cerevisiae* Pns1p does not actually transport choline [64], and its function is unknown [63]. In another study, two mammalian CTL-like proteins transported ethanolamine rather than choline [65]. This indicates that the CTL family of transporters may have a more diverse range of substrates than previously thought, and that the substrate(s) of the *Peronospora* CTLs will need to be identified in the future.

The RMX67305 gene encodes a putative NIPA-like protein, which is a selective magnesium transporter in mammals [66,67]. Magnesium, along with calcium, contributes to cross linking of the middle lamella in plants that makes this structure more resistant to the pectolytic enzymes produced by plant pathogens [68]. The *Peronospora* oomycete pathogens might actively remove magnesium from the plant middle lamella during infection by increasing the expression levels of their NIPA-like transporter-encoding genes. RQM14033 encodes a putative C4-dicarboxylate transport protein which has substantial homology (BLASTP E value = 9e-78) to the C4-dicarboxylate transporter from *Bacillus subtilis*, which is part of the DctA family of secondary transporters in bacteria [69]. The *in vitro* substrate specificity of the *B. subtilis* DctA transporter is limited to succinate, malate, fumarate, and oxaloacetate, which are C4-dicarboxylates of the Krebs cycle [70]. Similar *in vitro* experiments will need to be completed in the future to determine the substrate specificity of each *Peronospora* DctA-like transporter.

The RMX68746 gene encodes a putative UAA family transporter that is a nucleotide sugar transporter (NST); these transporters are critical for the movement of nucleotide sugars from the cytosol to the Golgi apparatus [71]. A phylogenetic analysis of 257 NST proteins showed that the substrate specificity of any NST if can be inferred from its primary sequence [71]. A BLASTP analysis indicated that the proteins encoded by the three *Peronospora* RMX68746 genes are part of the solute carrier family 35 (member B1 isoform 1), which corresponded to

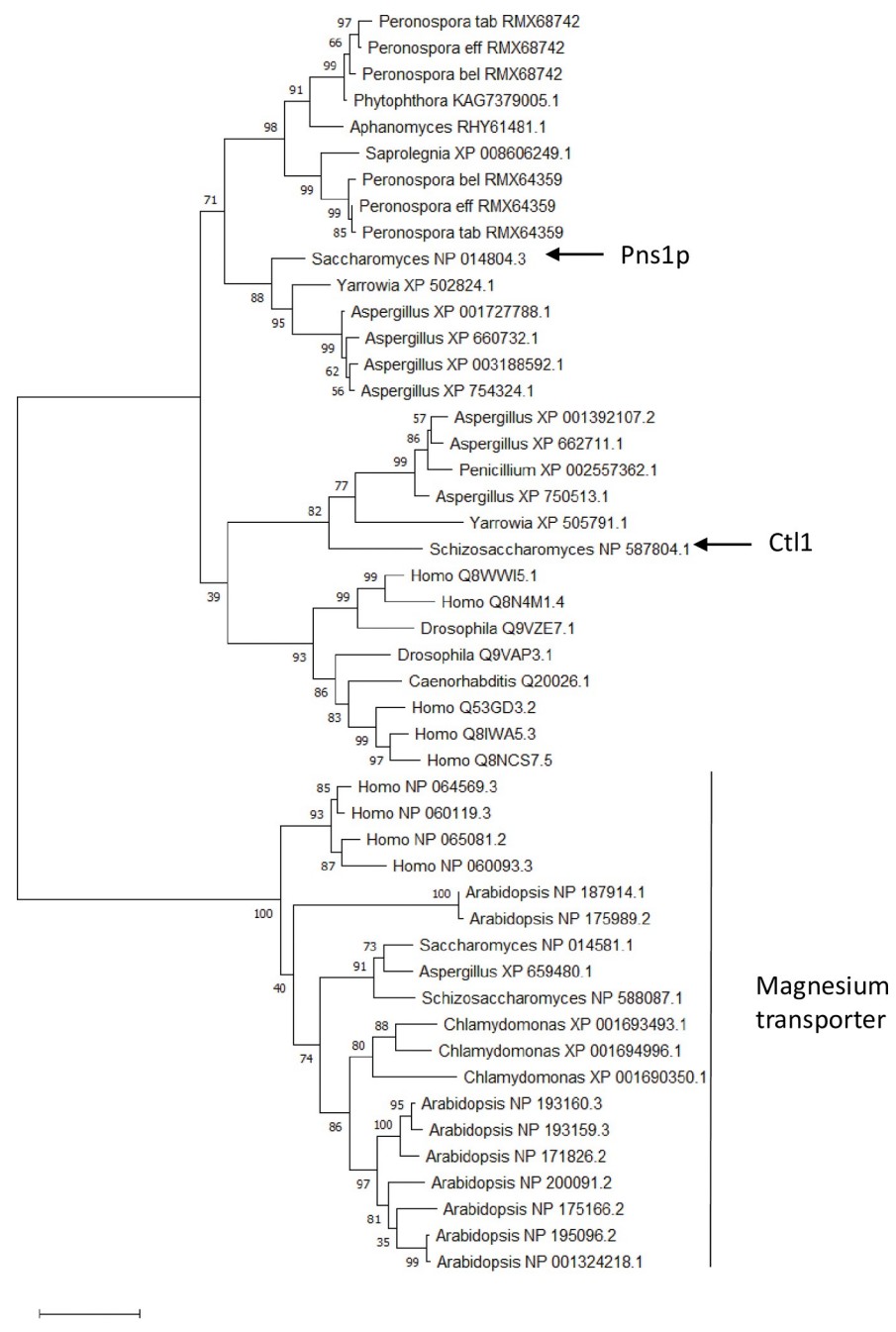

**Fig 4. Phylogenetic analysis of various CTL and magnesium transporters.** The maximum likelihood tree was constructed using the WAG+F model. The percentages of replicate trees in which the associated proteins clustered together in the bootstrap test (1,000 replicates) are shown adjacent to the nodes. The proteins directly to the left of the black line are the magnesium transporters, whereas the Ctl1 and Pns1p proteins are noted with arrows. The genus of the organism from which the protein originated and the Genbank number is given at the tip of each branch; a '_' follows each XP or NP. The scale bar indicates the number of protein substitutions per site.

clade F of the NST superfamily [71]. The proteins encoded by the three *Peronospora* RMX68746 genes clustered with all 15 clade F proteins [71] in a phylogenetic tree (S1 Fig). Clade F proteins transport uridine-5'-diphosphate (UDP)-galactose, UDP-glucose, and

adenosine 3'-phospho 5'-phosphosulfate [71]. This proposed substrate specificity of the *Peronospora* proteins encoded by the RMX68746 genes should be experimentally verified in the future.

## In-depth transporter protein analysis

The proteins encoded by the 16 transporter genes that were highly expressed in the three *Peronospora* organisms were examined in more detail (Table 4). Pairwise comparisons of each species with each other indicated a high degree of amino acid identity. Most of the *Peronospora* MC proteins were close to 300 amino acids in length, which is typical for these transporters [72]. The exception was protein A0A3M6VF52 from *P. effusa*, which was 586 amino acids long. Alignment of A0A3M6VF52 with the homologous proteins of *P. belbahrii* and *P. tabacina* showed that the homologous regions of the three proteins are in the C-terminus of A0A3M6VF52 (see S2 File). Mapping of the reads in the four *P. effusa* transcriptomes (7 DPI) to the *P. effusa* RMX65655 gene revealed that the reads primarily localized to the 3' end of A0A3M6VF52, which suggests that the *P. effusa* RMX65655 gene is not correctly annotated. The *P. effusa* RMX66310 gene encodes a 97-amino acid protein, but a protein from Genbank (UIZ22858.1) is 355 amino acids in length and is likely to be the authentic protein, because it shared 90% identity with Ptab2_001070 from *P. tabacina*.

There was substantial variability in amino acid sequence length among the three proteins (putative glucose transporters) in each *Peronospora* species encoded by the RQM16367 genes. GLUTs or glucose transporters should contain 12 membrane-spanning helices [73].

**Table 4. Properties of transporter proteins from three species of *Peronospora*.**

| MC[a] | *P. belbahrii*[b] | *P. effusa*[c] | *P. tabacina*[d] |
|---|---|---|---|
| RMX67221 | 309 [*Pb-Pt* 95][e] PBEL_08012 | 309 (6) [*Pe-Pb* 96] A0A3M6VMF9 | 309 (6) [*Pt-Pe* 98] Ptab2_013536 |
| RMX65655 | 345 [*Pb-Pt* 94] PBEL_07973 | 586 [*Pe-Pb* 97] A0A3M6VF52 | 345 [*Pt-Pe* 95] Ptab2_000589 |
| RMX67457 | 305 [*Pb-Pt* 89] PBEL_03868 | 274 [*Pe-Pb* 93] A0A3M6VM16 | 305 [*Pt-Pe* 96] Ptab2_004998 |
| RMX67668 | 288 [*Pb-Pt* 94] PBEL_00715 | 289 [*Pe-Pb* 92] A0A3M6VL63 | 288 [*Pt-Pe* 95] Ptab2_003968 |
| RMX63170 | 353 [*Pb-Pt* 82] PBEL_01172 | 336 [*Pe-Pb* 82] A0A3M6VFR0 | 338 [*Pt-Pe* 86] Ptab2_003919 |
| RMX66310 | 351 [*Pb-Pt* 77] PBEL_01047 | 355 [*Pe-Pb* 78] UIZ22858.1 | 354 [*Pt-Pe* 90] Ptab2_001070 |
| RMX69262 | 307 [*Pb-Pt* 92] PBEL_08758 | 306 [*Pe-Pb* 94] A0A3M6VSJ1 | 306 [*Pt-Pe* 95] Ptab2_027423 |
| MFS | | | |
| RMX65758 | 536 [*Pb-Pt* 89] PBEL_07211 | 536 [*Pe-Pb* 89] A0A3M6VFF3 | 536 [*Pt-Pe* 94] Ptab2_003158 |
| RMX65273 | 488 [*Pb-Pt* 88] PBEL_01212 | 494 [*Pe-Pb* 89] A0A3R7W5I9 | 487 [*Pt-Pe* 91] no gene name found |
| RQM16367 | 493 [*Pb-Pt* 80] PBEL_00785 | 595 [*Pe-Pb* 82] A0A3R7W609 | 262 [*Pt-Pe* 92] Ptab2_015719 |
| RMX67545 | 549 [*Pb-Pt* 93] PBEL_05539 | 551 [*Pe-Pb* 95] A0A3M6VKN9 | 551 [*Pt-Pe* 97] Ptab2_017636 |
| Others | | | |
| RMX64359 (CTL) | 479 [*Pb-Pt* 86] PBEL_00807 | 480 [*Pe-Pb* 87] A0A3M6VCI3 | 447[f] [*Pt-Pe* 93] Ptab2_000964 |
| RMX68742 (CTL) | 507 [*Pb-Pt* 75] PBEL_06413 | 505 [*Pe-Pb* 82] A0A3M6VR13 | 478[f] [*Pt-Pe* 84] Ptab2_000106 |
| RMX67305 (DMT) | 475 [*Pb-Pt* 80] PBEL_01371 | 460 [*Pe-Pb* 82] A0A3M6VK02 | 460 [*Pt-Pe* 84] Ptab2_022810 |
| RQM14033 (DAACS) | 458 [*Pb-Pt* 83] PBEL_07339 | 464 [*Pe-Pb* 84] A0A425CAG7 | 461 [*Pt-Pe* 91] Ptab2_014153 |
| RMX68746 (DMT) | 327 [*Pb-Pt* 85] PBEL_06418 | 327 [*Pe-Pb* 90] A0A3M6VT22 | 327[f] [*Pt-Pe* 92] Ptab2_009561 |

[a]See Table 1 for transporter abbreviations in the first column.

[b]The protein designations are provided from the European Nucleotide Archive accession ERZ1462836.

[c]The proteins are from the InterPro website or Genbank.

[d]The protein designations are from GenBank accession GCA_002099245.

[e]In each cell, the first number is the length of the protein encoded by the gene in each species. A pairwise value of percent sequence identity (calculated by Clustal Omega) is provided in the brackets where *Pb* is *P. belbahrii*, *Pe* is *P. effusa* and *Pt* is *P. tabacina*.

[f]The gene encoding this protein contained heterozygous nucleotides; see the text for how the pairwise comparisons were made.

A0A3R7W609 was predicted to contain 11, 14, and 11 transmembrane helices whereas PBEL_00785 was predicted to contain 11, 12, and 9 transmembrane helices according to the transmembrane helix predictor programs HMMTOP [38], deep TMHMM [39] and SPLIT 4.0 [40], respectively. The variability in these predictor programs indicates that more detailed protein structural analysis needs to be performed. However, it is likely that the actual coding sequence for Ptab2_015719, the protein from *P. tabacina*, is incomplete.

It should be noted that the genes Ptab2_000964, Ptab2_000106, and Ptab2_009561 all contained heterozygous nucleotide sites. The encoded amino acids at these heterozygous sites were arbitrarily chosen to be a glycine (G) so that it would not match the amino acid at that position in *P. belbahrii* or *P. effusa*; if adding a G resulted in a perfect match at the *P. tabacina* heterozygous site among all three *Peronospora* species, a different amino acid was randomly added at that site to purposefully cause mismatches among the three *Peronospora* proteins. Even with these intentional mismatches included in the proteins encoded by genes with heterozygous nucleotide sites, the transporter proteins in the three *Peronospora* were highly similar (75–92% identical).

Protein modelling was completed for the ADP/ATP translocase, *P. effusa* A0A3M6VMF9, the mitochondrial phosphate carrier protein, *P. belbahrii* PBEL_07973, and one glucose transporter, *P. effusa* A0A3R7W609 (details of the models are in S3 File). These specific proteins were chosen for modelling because of their high levels of expression and presumed important roles in all three *Peronospora* species during host infection. Each *Peronospora* protein model was aligned to a protein model with the same putative function in a mammal (Fig 5). The ADP/ATP translocase pair had 58% sequence identity and a TM-score of 0.69 if normalized to the length of the *P. effusa* protein; TM-scores range from 0 to 1, and scores above 0.5 indicate that the proteins have similar folding patterns [74]. The mitochondrial phosphate carrier and glucose transporter pairs had 48% and 31% sequence identity and TM-scores of 0.77 and 0.83, respectively (both TM-scores were normalized to the length of the *Peronospora* protein).

ADP/ATP translocases have been extensively studied. Yeast and bovine ADP/ATP translocases share almost 50% amino acid sequence identity [50]. The bovine protein used in the structural comparison shown in Fig 5A, 1OKC, was crystallized with an inhibitor, carboxyatractyloside [50]. *P. effusa* A0A3M6VMF9 and 1OKC share 67% amino acid identity as determined by the EMBOSS needle alignment algorithm. The RRRMMM motif in the bovine ADP/ATP translocase (which is also present in the three *Peronospora* translocases) was postulated to be a two-way switch which regulates the nucleotide binding stoichiometry [50].

The mammalian phosphate carrier, also known as SLC25A3, has two isoforms, A and B, that differ by 13 amino acids between residues 54 and 80 [75]. The structure of SLC25A3 isoform B from humans was used for comparison with *P. belbahrii* PBEL_07973 (Fig 5B). Bovine SLC25A3 isoform B was shown to have a phosphate transport rate that was ~3-fold higher than the phosphate transport rate of bovine SLC25A3 isoform A in liposomes [76]. Human SLC25A3-A and SLC25A3-B transported phosphate in *Lactococcus lactis*, as the transformed cells were unable to grow in 1.6 mM arsenate, a toxic mimetic of phosphate [75]. Surprisingly, human SLC25A3-A and SLC25A3-B also exhibited the ability to transport copper, because the *L. lactis* transformants were unable to grow in the presence of 100 µM silver, which is an indirect method to demonstrate copper transport [75]. The PIC2 protein transports phosphate and copper in *Saccharomyces cerevisiae*, but MIR1 can only transport phosphate in this organism; the different substrate specificities of these two mitochondrial carriers suggests that they evolved from an ancient gene duplication [77]. There are PIC2-like and MIR1-like proteins present throughout eukaryotic lineages [77]. Based on their clustering pattern in a phylogenetic tree (S2 Fig), the *Peronospora* phosphate carriers identified in this study (PBEL_07973, A0A3M6VF52, and Ptab2_000589; Table 4) are PIC2-like proteins and could possibly transport both phosphate and copper. The *P. effusa* genome is not likely to contain any MIR1-like

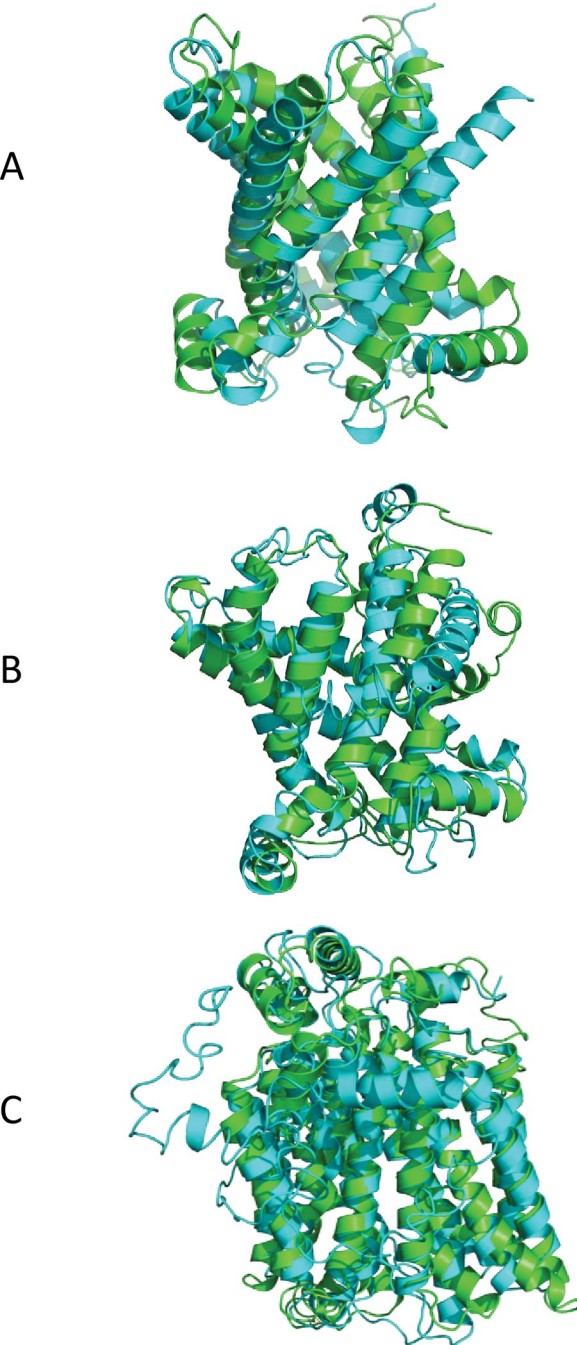

**Fig 5. Aligned structures of the modeled proteins.** A, A0A3M6VMF9 from *P. effusa* (green) and 1OKC (PDB) from *Bos taurus* (cyan); B, PBEL_07973 from *P. belbahrii* (cyan) and NP_002626.1 from *Homo sapiens* (green); C, A0A3R7W609 from *P. effusa* (cyan) and XP_011511389.1 from *Homo sapiens* (green).

proteins because the top BLASTP hit using several oomycete MIR1-like proteins as queries at the NCBI server was the PIC2-like protein A0A3M6VF52 (data not shown); however, more complete annotation of the *Peronospora* genomes may uncover some MIR1-like proteins.

In the last structural comparison (Fig 5C), the human glucose transporter GLUT2 (XP_011511389.1) and *P. effusa* A0A3R7W609 were found to share 24% amino acid sequence

identity using the EMBOSS needle alignment algorithm [42] at the following Internet server [43]. GLUT2 can also transport galactose, mannose, fructose, and glucosamine [78]. This broad substrate specificity of GLUT2 suggests that substrate testing should be performed for all the *Peronospora* carbohydrate transporters in the future.

## Importance of transporters in pathogen nutrition, virulence, and defense

The majority of the highly expressed transporters in this study (Fig 1) contribute to the movement of essential molecules across the inner mitochondrial membrane [79]. In addition, we found that many MFS transporter genes were well expressed during infection. Some *Peronospora* MFS transporters likely contribute to the movement of carbohydrate(s) from the host to the pathogen, which is vital for pathogen growth within the plant, but other *Peronospora* MFS transporters could be involved in the movement of plant defensive products, or man-made fungicides, out of the pathogen. An MFS transporter in *Botrytis cinerea* promotes tolerance to isothiocyanate, a breakdown product of glucosinolates, which are natural defense compounds produced by species in the Brassicaceae [80]. The authors of the previously mentioned study on *B. cinerea* speculated that the large number of MFS transporters in that pathogen are utilized to counteract antimicrobial compounds produced by plants during infection. There are also some reports that microbial MFS transporters secrete pathogenicity factors or toxins into plant tissues [81], and these MFS transporters may also protect the pathogens themselves from deleterious levels of their own toxins [82,83]. On another note, some eukaryotic organisms develop multidrug resistance, as well as fungicide resistance, through the expression of ABC and MFS transporters that export the fungicide out of the cell [84]. For example, mutation of an MFS transporter in *Alternaria alternata* increased the sensitivity of the mutant to several fungicides [85]. Disruption of *MgMfs1* in *Mycosphaerella graminicola* increased the mutant's sensitivity to several strobilurin fungicides [86]. The substrate specificities of the many *Peronospora* MFS transporters need to be identified to determine whether each transporter plays role in pathogen protection or pathogenicity.

This study identified the genes that encode several *Peronospora* transporters that are likely to be crucial for growth and colonization of their plant host. On the one hand, whereas it is relatively straightforward to characterize the gene expression patterns of pathogen transporter genes during infection, the bottleneck in further knowledge of these membrane proteins will be the identification of transporter substrates, which is usually done with artificial membranes, or in yeast, and recombinant expression of the transporter protein. On the other hand, the identification of these crucial transporter genes in *Peronospora* pathogens could be helpful for designing crop protection products against pathogens. The addition of small double-stranded RNA molecules targeting a downy mildew pathogen's cellulose synthase gene to the spores of *Hyaloperonospora arabidopsidis* inhibited infection of its host plant [87]. It is conceivable that downy mildew pathogen infection of basil, spinach, or tobacco could be inhibited by targeting one of the 16 well-expressed transporter genes identified in this study. The use of gene silencing RNA molecules for plant protection is still relatively untested in agricultural settings, but the concept is being examined for control of several fungal and oomycete plant pathogens [88,89]. New control strategies need to be developed in the near future because downy mildew causing pathogens can evade current control technologies over time.

## Supporting information

**S1 Fig. Phylogenetic tree of Clade F NST proteins.** The neighbor-joining phylogenetic tree was constructed using the JTT model. The percentage of replicate trees in which the associated proteins clustered together in the bootstrap test (1000 replicates) are posted next to the

branches. The F and I proteins, listed with their Uniprot accession numbers, formed two separate clades in the tree. The bar indicates the number of protein substitutions per site.
(PPTX)

**S2 Fig. Phylogenetic tree of PIC2-like and MIR-like proteins.** The maximum likelihood tree was constructed using the LG model. The percentage of replicate trees in which the associated proteins clustered together in the bootstrap test (1000 replicates) are posted next to the branches. The proteins in the red box are putative PIC2-like; the proteins in the blue box are putative MIR1-like transporters. The genus of the organism from which the protein originated and the Genbank number is listed on each branch; a '_' follows each XP or NP. The bar indicates the number of protein substitutions per site.
(PPTX)

**S1 Table. Quality metrics for tobacco leaf transcriptome samples infected with *P. tabacina*.**
(XLSX)

**S2 Table. Motif scores for all the putative transporters identified in the three *Peronospora* species.**
(XLSX)

**S3 Table. Gene expression values for *P. effusa*.**
(XLSX)

**S4 Table. Gene expression values for *P. belbahrii*.**
(XLSX)

**S5 Table. Gene expression values for *P. tabacina*.**
(XLSX)

**S1 File. DNA sequences of housekeeping genes of all three *Peronospora* species.**
(TXT)

**S2 File. Multiple sequence alignment of the phosphate carrier proteins in the three *Peronospora* species.**
(DOCX)

**S3 File. Confidence tests for the modeled transporter proteins.**
(DOCX)

## Acknowledgments

We thank Mark Doehring for excellent technical assistance and Shyam Kandel for reviewing the draft manuscript. The mention of firm (company) names or trade products does not imply that they are endorsed or recommended by the USDA over other firms (companies) or similar products not mentioned. USDA is an equal opportunity provider and employer.

## Author Contributions

**Conceptualization:** Eric T. Johnson, David Zaitlin.

**Formal analysis:** Eric T. Johnson, Rebecca Lyon, Abdul Burhan Khan, Mohammad Aman Jairajpuri.

**Methodology:** Eric T. Johnson, Rebecca Lyon, David Zaitlin, Abdul Burhan Khan, Mohammad Aman Jairajpuri.

**Supervision:** Eric T. Johnson, Mohammad Aman Jairajpuri.

**Writing – original draft:** Eric T. Johnson, Abdul Burhan Khan.

**Writing – review & editing:** Eric T. Johnson, Rebecca Lyon, David Zaitlin, Abdul Burhan Khan, Mohammad Aman Jairajpuri.

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
