## [Decision Letter · Decision Letter 0]

28 Nov 2022

PONE-D-22-25502A comparison of transporter gene expression in three species of Peronospora plant pathogens during host infectionPLOS ONE

Dear Dr. Johnson,

Thank you for submitting your manuscript to PLOS ONE. You may confirm that both reviewers acknowledged the quality of your study, and I agree on them, but raised very constructively a number of important issues that need to be addressed before publication. Therefore, we invite you to submit a revised version of the manuscript that addresses the points raised during the review process.

We look forward to receiving your revised manuscript.

Kind regards,

Hernâni Gerós, PhD

Academic Editor

PLOS ONE

Journal Requirements:

2. Please expand the acronym “USDA ARS” (as indicated in your financial disclosure) so that it states the name of your funders in full.

4. Please upload a new copy of Figures 1, 2, 3 and 4 as the detail is not clear. Please follow the link for more information: " ext-link-type="uri" xlink:type="simple">https://blogs.plos.org/plos/2019/06/looking-good-tips-for-creating-your-plos-figures-graphics/"
https://blogs.plos.org/plos/2019/06/looking-good-tips-for-creating-your-plos-figures-graphics/

Reviewers' comments:

Reviewer's Responses to Questions

**Comments to the Author**

1. Is the manuscript technically sound, and do the data support the conclusions?

Reviewer #1: Partly

Reviewer #2: Yes

2. Has the statistical analysis been performed appropriately and rigorously? 

Reviewer #1: No

Reviewer #2: Yes

3. Have the authors made all data underlying the findings in their manuscript fully available?

Reviewer #1: Yes

Reviewer #2: Yes

4. Is the manuscript presented in an intelligible fashion and written in standard English?

Reviewer #1: No

Reviewer #2: Yes

5. Review Comments to the Author

Reviewer #1: Dear Authors,

Your Plos1 report on Peronospora transcriptomics was extended for my peer-review. In it, you generate RNAseq data for P. tabacina and re-analyze similar data from two related pathogens. Your study is a report on bioinformatic analyses with virtually no wet-lab studies beyond these required for the generation of RNAseq data. As such, this leaves me wanting for more rigorous research, to substantiate your conclusions. Writing of this report is generally acceptable, but I flag below numerous issues, with those indicated as [MAJOR] being of particular importance, in my opinion.

l.22 onto their hosts {specify}

l.22-24 How measured?

l/43 vs. l.50 [MAJOR] Phytopathologically, one cannot control/manage a disease. Disease is the result (signs and symptoms) of compatible interaction between a pathogen and a (susceptible) plant host. Phrases as in l.50 or "resistance to downy mildew" l.50-51 are mental shortcuts, not to be used in technical, precise, scientific writing. Furthermore, Authors DO use occasionally the correct phrasing: l.56 "pathogen is controlled". Please unify/correct throughout.

52 resistant - resistance

60 while - whereas {correct throughout}

71 while - Although

87 'that encode... carbohydrate' please rephrase for clarity/meaning

99 [MAJOR] Be specific. This paragraph and those that follow actually only des"Protein alignments" - "Sequence alignmentrs"cribe P. tabacina processing.

122 Describe briefly the RNA isolation/quality checks; do not make readers look in other sources.

130-131 CITE if published

140 see comment above

152 [MAJOR] Describe what was used as house-keeping gene(s) incl. GenBank seq numbers and citations, if published. Also add, what sample(s) served as baseline.

154 Proteins - Amino acid sequences (if that was the intended meaning). "Protein alignments" - Sequence alignments

[MAJOR] Details of this analysis missing : What was the mutation model/matrix used, and how was it chosen? Was parsimony correction implemented? What was the outgroup / root and why? How many bootstraps used? What was the software (add version and citation)?

161 {in the range of}

162 high{er}

166-167 [MAJOR] Please provide more substantiation for this focus.

173-174 was not statistically significant - lacked statistical support

181 significant number (How was the significance of this result assessed? This number is comparable to MFS)

185 measured - assessed/analyzed(?)

188-189 significant expression {Again, the rationale/test data for the significance claim are missing; What was the significance threshold? How established?}

190 "Many more..." please append with the respective data point (Table/Figure)

190-191 [MAJOR] This sentence rather indicates higher expression levels than more genes expressed. What other analyses / results can support the original claim?

200 "The same transporter genes were measured in RNA samples" Imprecise. Please detail what was done and how.

210 {expression levels of} the PBEL_08012... [drop the TPM]

225 and 234 [MAJOR] the top 50 genes / 45 genes {By what feature? Same feature in both cases?}

236 Consider adding more broad-scale context, as available in the following papers: 10.1093/gbe/evz048 ; 10.1016/j.fgb.2006.07.005 ; 10.1016/bs.adgen.2020.03.001 ; 10.1094/PHYTO-03-21-0092-R ; 10.1146/annurev-phyto-102313-050056 ; 10.1080/07352689.2018.1530848 ; 10.1111/nph.16092 ; 10.1094/phyto-05-15-0127-rvw ; 10.1101/2020.02.12.941195

243 (30) Other great resources to add context are: 10.1111/mpp.12263 ; 10.1111/mpp.12260

291 Curiously (?) Consider rephrasing

297 CTL [Abbreviation not introduced]

312 [MAJOR] substance {A major offence to cell biology. Consider replacing with "structure" or similar}

335 well - highly

340-345 [MAJOR] Methodological details of all that are missing in MM.

353 [MAJOR] HMMTOP and TMHMM are missing from MM.

372 [MAJOR] TM-align details are missing from MM.

382-383 [MAJOR] Details missing from MM.

406 discover - uncover

414 Most - Majority

417 carbohydrate{s}

434 [MAJOR] Consider discussing the context of P.tabacina fungicide resistance, e.g. 10.1094/PHYTO-03-21-0092-R ; 10.3390/agronomy5040555 and the older, descriptive studies

442 [MAJOR] downy mildews - pathogens

453;454 firm - company

459 But (?)

462 KhanC (?)

Reviewer #2: Dear editor and authors,

The authors developed in this study a comparison of gene expression of transporter proteins of three different Peronospora species during infection. The manuscript is simple, clear, well structured and written. For the study, they performed the infection experiments for one species and used two different public databases of published experiments. Therefore, this study is partially based on original scientific research but the analysis of the public databases is interesting. The standard of the experiments is of high degree and the methods are sufficiently detailed, as well as the statistical analysis. The conclusions are based in the results and are well presented.

In the next line I present a series of recommendations and questions to debate with the authors.

• Line 26: This type of discussion is not adequate for the abstract.

• Why the authors blasted the putative transporters o P. belbahrii and P. tabacina from the P. effusa ones and not from the paper that you identified the P. effusa ones?

• What is the rational of study these 3 pathogen species?

• Can the authors explain the discrepancies regarding the days after inoculation of the samples?

• Are not other databases using P. effusa pathogenesis that can be used because of first point of infection?

• I cannot understand the value of the bars graphs presented in figure 1 without the names of the genes.

• Line 197: “In addition, the TPM levels of two MFS, two MC and one Amino Acid/Auxin Permease (AAAP) gene were close to the same TPM levels of the housekeeping genes tubulin B and pyruvate 199 dehydrogenase alpha subunit in P. effusa at 7 DPI (Table 2).” Why is this relevant?

• The gene code presented in table 1 is the “same” for homologous genes? Is confusing the nomenclature in the text differs from the table. Maybe you can opt to introduce the specie gene name and its homologue.

• Line 215: “The TPM level of the MC gene PBEL_00715, as well as that of the actin gene, decreased from 3 216 to 6 DPI.” Why is this line relevant? Are these the only genes that decrease from 3 to 6 dpi?

• Why did you choose Phytophthora infestans for comparison?

• In the section “Possible functions for the highly-expressed Peronospora transporter genes” the authors need to be careful when referring to a gene which function is characterized or is simply a approximation by homology. Please be clear in all your discussion. “RMX69262 encodes a putative 2-oxoglutarate/malate carrier protein, also known as the mitochondrial dicarboxylate-tricarboxylate carrier (DTC) protein (49). The RMX69262 protein from P. effusa shares 38% identity with A. thaliana DTC (At5g19760, BLASTP E value = 2e-67), which specifically transports protonated citrate and unprotonated malate (49); DTCs can also transport oxaloacetate, oxoglutarate, isocitrate cis-aconitate, and trans-aconitate (49).” This is a good example that you can replicate along all this part of the discussion.

• In the comparison to Phytophthora infestans, why are you comparing your set of “late” infection time point to 2 DPI? The necrotrophic phase begins in P. infestans at 3 dpi? (You do not have this info in the methods).

• How the expression numbers in P. infestans study that you present in table 2 compare to your study?

• Line 256: “The results of a previous study indicated that both ADP/ATP translocase and phosphate carrier protein levels were high in A. thaliana (43)…” In what condition?...” suggesting that these proteins were part of a metabolic interaction (44).” What kind of metabolic interaction?

Minor changes:

Line 102 and 117: Specify ~100.000 spores/ mL.

Line 228: Typo. Is 2 or 5 DPI.

6. PLOS authors have the option to publish the peer review history of their article (what does this mean?). If published, this will include your full peer review and any attached files.

Reviewer #1: No

Reviewer #2: No

---

## [Author Response · Author response to Decision Letter 0]

17 Jan 2023

Reviewer #1: Dear Authors,

Your Plos1 report on Peronospora transcriptomics was extended for my peer-review. In it, you generate RNAseq data for P. tabacina and re-analyze similar data from two related pathogens. Your study is a report on bioinformatic analyses with virtually no wet-lab studies beyond these required for the generation of RNAseq data. As such, this leaves me wanting for more rigorous research, to substantiate your conclusions. Writing of this report is generally acceptable, but I flag below numerous issues, with those indicated as [MAJOR] being of particular importance, in my opinion.

l.22 onto their hosts {specify} 

Changed to: onto leaves of their hosts. The hosts are specified in the fourth sentence of the abstract, so we do not think it is necessary to name the hosts again.

l.22-24 How measured? 

Changed to: measured using RNA sequencing.

l/43 vs. l.50 [MAJOR] Phytopathologically, one cannot control/manage a disease. Disease is the result (signs and symptoms) of compatible interaction between a pathogen and a (susceptible) plant host. Phrases as in l.50 or "resistance to downy mildew" l.50-51 are mental shortcuts, not to be used in technical, precise, scientific writing. Furthermore, Authors DO use occasionally the correct phrasing: l.56 "pathogen is controlled". Please unify/correct throughout. 

Resistance to downy mildew, or similar phrase, changed throughout the manuscript.

52 resistant - resistance

 changed as recommended

60 while - whereas {correct throughout} 

corrected throughout the manuscript 

71 while - Although 

changed as recommended

87 'that encode... carbohydrate' please rephrase for clarity/meaning

 this sentence was edited

99 [MAJOR] Be specific. This paragraph and those that follow actually only describe P. tabacina processing.

The subtitle on lines 99 and 114 were modified to be more specific

122 Describe briefly the RNA isolation/quality checks; do not make readers look in other sources. 

A brief description of extraction of RNA from leaves was added.

130-131 CITE if published

 a citation was added

140 see comment above

 a citation was added

152 [MAJOR] Describe what was used as house-keeping gene(s) incl. GenBank seq numbers and citations, if published.

An additional supplementary file was added that contained all the sequences of the housekeeping genes; for this reason GenBank sequence numbers were not included in the supplementary file. 

Also add, what sample(s) served as baseline.

No samples served as a baseline in this study; gene expression in each pathogen at each timepoint was analyzed independently. 

154 Proteins - Amino acid sequences (if that was the intended meaning). "Protein alignments" - Sequence alignments the. 

Changed as recommended.

[MAJOR] Details of this analysis missing : What was the mutation model/matrix used, and how was it chosen? Was parsimony correction implemented? What was the outgroup / root and why? How many bootstraps used?

 These details are written in the figure legends.

 What was the software (add version and citation)? The software program MEGA version 10 was listed and cited in this section of the Materials and Methods.

161 {in the range of} Changed as recommended.

162 high{er} if Changed as recommended.

166-167 [MAJOR] Please provide more substantiation for this focus.

We think that the last paragraph of the introduction discusses why we chose to analyze gene expression of transporters.

173-174 was not statistically significant - lacked statistical support. 

Changed as recommended.

181 significant number (How was the significance of this result assessed? This number is comparable to MFS).

We agree that the use of ‘significant’ is not appropriate; therefore this word was removed. 

185 measured - assessed/analyzed(?).

We substituted ‘analyzed’ for ‘measured’.

188-189 significant expression {Again, the rationale/test data for the significance claim are missing; What was the significance threshold? How established?}. We substituted ‘enhanced’ for ‘significant’.

190 "Many more..." please append with the respective data point (Table/Figure).

The data are in Supplementary File 3, which was noted in the sentence.

190-191 [MAJOR] This sentence rather indicates higher expression levels than more genes expressed. What other analyses / results can support the original claim? This sentence was edited for more clarity.

200 "The same transporter genes were measured in RNA samples" Imprecise. Please detail what was done and how.

More detail was added to this sentence; the ‘how’ is described in the materials and methods section.

210 {expression levels of} the PBEL_08012... [drop the TPM]. Changed as suggested.

225 and 234 [MAJOR] the top 50 genes / 45 genes {By what feature? Same feature in both cases?}.

The sentence describing the top 50 and top 45 most highly expressed genes was edited for clarity. 

236 Consider adding more broad-scale context, as available in the following papers: 10.1093/gbe/evz048 ; 10.1016/j.fgb.2006.07.005 ; 10.1016/bs.adgen.2020.03.001 ; 10.1094/PHYTO-03-21-0092-R ; 10.1146/annurev-phyto-102313-050056 ; 10.1080/07352689.2018.1530848 ; 10.1111/nph.16092 ; 10.1094/phyto-05-15-0127-rvw ; 10.1101/2020.02.12.941195.

A few more sentences were added at this point to provide additional context.

243 (30) Other great resources to add context are: 10.1111/mpp.12263 ; 10.1111/mpp.12260.

A few sentences were added to describe the biotrophic phase of P. infestans.

291 Curiously (?) Consider rephrasing. ‘Curiously’ was removed.

297 CTL [Abbreviation not introduced]. Abbreviation was defined

312 [MAJOR] substance {A major offence to cell biology. Consider replacing with "structure" or similar}. ‘Structure’ was used instead of ‘substance’.

335 well - highly. Changed as suggested.

340-345 [MAJOR] Methodological details of all that are missing in MM.

We used Clustal omega for the proteins sequence alignment; this analysis was described in the materials and methods. We changed the manuscript text at line 340 to make this more clear. We added more details about the mapping analysis to the materials and methods section.

353 [MAJOR] HMMTOP and TMHMM are missing from MM. Details of this analysis, including that for SPLIT 4.0, were added to the materials and methods.

372 [MAJOR] TM-align details are missing from MM. Details of this analysis were added to the materials and methods.

382-383 [MAJOR] Details missing from MM. Details of this analysis were added to the materials and methods.

406 discover - uncover. Changed as suggested.

414 Most - Majority. Changed as suggested.

417 carbohydrate{s}. Changed as suggested.

434 [MAJOR] Consider discussing the context of P.tabacina fungicide resistance, e.g. 10.1094/PHYTO-03-21-0092-R ; 10.3390/agronomy5040555 and the older, descriptive studies.

We think that a discussion about fungicide resistance in P. tabacina is moving away from the topic of the manuscript.

442 [MAJOR] downy mildews - pathogens. Changed as suggested.

453;454 firm - company. Added (companies) to the sentence.

459 But (?). Removed ‘But’.

462 KhanC (?). Removed the superscript.

Reviewer #2: Dear editor and authors,

The authors developed in this study a comparison of gene expression of transporter proteins of three different Peronospora species during infection. The manuscript is simple, clear, well structured and written. For the study, they performed the infection experiments for one species and used two different public databases of published experiments. Therefore, this study is partially based on original scientific research but the analysis of the public databases is interesting. The standard of the experiments is of high degree and the methods are sufficiently detailed, as well as the statistical analysis. The conclusions are based in the results and are well presented.

In the next line I present a series of recommendations and questions to debate with the authors.

• Line 26: This type of discussion is not adequate for the abstract. This specific line is a summary of our data in two different sentences. These data are mentioned in more detail in the discussion section. 

• Why the authors blasted the putative transporters of P. belbahrii and P. tabacina from the P. effusa ones and not from the paper that you identified the P. effusa ones? Kandel et al. did transcriptome analysis of spinach infected with P. effusa, however they did not identify any putative transporter genes in P. effusa. The transporter proteins of P. effusa were predicted from its genome sequence and downloaded from the InterPro website. These putative transporter proteins were checked for a statistically significant transporter motif and then the quality checked protein sequences were used to identify the putative transporters of P. belbahrii and P. tabacina.

• What is the rational of study these 3 pathogen species? Downy mildew disease causes major losses on economically important plants, such as those in this study; it is important to find out more detail about how these pathogens infect their hosts. In addition, we thought it would be of interest to analyze transporter gene expression in these three pathosystems to see if there are similarities or differences in the most highly expressed transporter genes during host infection.

• Can the authors explain the discrepancies regarding the days after inoculation of the samples? The studies of the basil, spinach and tobacco were independently completed. However, we felt that we could analyze this data together in this paper since the early timepoint (2 or 3 DPI) or the late timepoint (5, 6, or 7 DPI) only deviated one day or le ss than one day.

• Are not other databases using P. effusa pathogenesis that can be used because of first point of infection? We are not aware of additional P. effusa-spinach transcriptome data that is publicly available. In other words, we could only find one set of P. effusa-spinach transcriptome data.

• I cannot understand the value of the bars graphs presented in figure 1 without the names of the genes. We think it would be difficult to include the names of 50 genes on the horizontal axis. Readers would be able to find the gene name using the supplementary tables S3-S5.

• Line 197: “In addition, the TPM levels of two MFS, two MC and one Amino Acid/Auxin Permease (AAAP) gene were close to the same TPM levels of the housekeeping genes tubulin B and pyruvate 199 dehydrogenase alpha subunit in P. effusa at 7 DPI (Table 2).” Why is this relevant? We think this is relevant because we identified two transporter genes with expression levels close to the expression levels of important housekeeping genes. This suggests that these two transporter genes have an important function in the pathogen.

• The gene code presented in table 1 is the “same” for homologous genes? Is confusing the nomenclature in the text differs from the table. Maybe you can opt to introduce the specie gene name and its homologue. Admittedly it was difficult to identify a system for naming homologous genes. We settled on using the P. effusa RMX or RQM number as the homologous gene identifier in each Peronospora species. Readers can notice that the RMX/RQM naming system is used in Tables 2-4 for cross comparison. We think that we went through the Tables sequentially in the manuscript and introduce protein designations in the text (which were species specific) as needed. Moreover, Reviewer #1 did not mention a problem with our homologous gene identifiers.

• Line 215: “The TPM level of the MC gene PBEL_00715, as well as that of the actin gene, decreased from 3 216 to 6 DPI.” Why is this line relevant? Are these the only genes that decrease from 3 to 6 dpi? We thought it was interesting to find the expression (TPM level) of a transporter gene to decrease as the pathogen grew throughout the host. This is why we mentioned it.

• Why did you choose Phytophthora infestans for comparison? We chose to compare our data with Phy. infestans because there was a paper measuring transporter gene expression during infection on tomato plants. We were curious to find any similarities in the expression patterns of the transporter genes. Phy. infestans is an oomycete but not an obligate plant pathogen. 

• In the section “Possible functions for the highly-expressed Peronospora transporter genes” the authors need to be careful when referring to a gene which function is characterized or is simply a approximation by homology. Please be clear in all your discussion. “RMX69262 encodes a putative 2-oxoglutarate/malate carrier protein, also known as the mitochondrial dicarboxylate-tricarboxylate carrier (DTC) protein (49). The RMX69262 protein from P. effusa shares 38% identity with A. thaliana DTC (At5g19760, BLASTP E value = 2e-67), which specifically transports protonated citrate and unprotonated malate (49); DTCs can also transport oxaloacetate, oxoglutarate, isocitrate cis-aconitate, and trans-aconitate (49).” This is a good example that you can replicate along all this part of the discussion. 

We have gone through this section of the manuscript to make sure that we state a putative or probable function. Verification of the possible function will need to be pursued in later studies.

• In the comparison to Phytophthora infestans, why are you comparing your set of “late” infection time point to 2 DPI? The necrotrophic phase begins in P. infestans at 3 dpi? (You do not have this info in the methods). In the paper by Abrahamian et al. 2016, the P. infestans should be at the biotrophic at 2 DPI in tomato leaves. This was stated in the results and discussion section, in the first paragraph below the subtitle 'Possible functions for the highly-expressed Peronospora transporter genes'. 

• How the expression numbers in P. infestans study that you present in table 2 compare to your study? The expression value and ranking for each P. infestans gene is listed in the far right column of Table 3. It would be difficult to compare expression values from the P. infestans and our study directly, and this is why we utilized the ranking as a means of comparison. This was stated in the results and discussion section, in the first paragraph below the subtitle 'Possible functions for the highly-expressed Peronospora transporter genes'.

• Line 256: “The results of a previous study indicated that both ADP/ATP translocase and phosphate carrier protein levels were high in A. thaliana (43)…” In what condition?.. This sentence was modified to state that the ADP/ATP translocase and the phosphate carrier were the most abundant proteins in the inner mitochondrial membrane of Arabidopsis plants according to Fuchs et al. 2020. “suggesting that these proteins were part of a metabolic interaction (44).” What kind of metabolic interaction? The phrase about the metabolic interaction was removed, as Kunji et al. suggest that the ADP/ATP translocase has no functional interactions with any other proteins (https://doi.org/10.1016/j.bbamcr.2016.03.015).

Minor changes:

Line 102 and 117: Specify ~100.000 spores/ mL. Changed as suggested. 

Line 228: Typo. Is 2 or 5 DPI. Changed to 5 DPI.

---

## [Decision Letter · Decision Letter 1]

28 Feb 2023

PONE-D-22-25502R1A comparison of transporter gene expression in three species of Peronospora plant pathogens during host infectionPLOS ONE

Dear Dr. Johnson,

Thank you for submitting your manuscript to PLOS ONE. After careful consideration, we feel that it has merit but does not fully meet PLOS ONE’s publication criteria as it currently stands. Therefore, we invite you to submit a revised version of the manuscript that addresses the points raised during the review process. Please, improve the abstract, as reviewer #2 suggests, by removing of modifying the following two sentences "Several of the 16 highly-expressed transporter genes identified in this study were also highly-expressed in Phytophthora infestans during biotrophic infection of  tomato leaves as measured in a previously published study. Comparisons to other studied transporters indicated that most of the 16 highly-expressed transporter genes have a putative function that will need to be confirmed in future experiments." Also, do not forget to clarify figure 1 to make it informative, as suggested by the reviewer #2. Please, also carefully address the remaining remarks of the reviewer #1. Please submit your revised manuscript by Apr 14 2023 11:59PM. If you will need more time than this to complete your revisions, please reply to this message or contact the journal office at plosone@plos.org. Please include the following items when submitting your revised manuscript:A rebuttal letter that responds to each point raised by the academic editor and reviewer(s). You should upload this letter as a separate file labeled 'Response to Reviewers'.A marked-up copy of your manuscript that highlights changes made to the original version. You should upload this as a separate file labeled 'Revised Manuscript with Track Changes'.An unmarked version of your revised paper without tracked changes. You should upload this as a separate file labeled 'Manuscript'.If applicable, we recommend that you deposit your laboratory protocols in protocols.io to enhance the reproducibility of your results. Protocols.io assigns your protocol its own identifier (DOI) so that it can be cited independently in the future. For instructions see: https://journals.plos.org/plosone/s/submission-guidelines#loc-laboratory-protocols. Additionally, PLOS ONE offers an option for publishing peer-reviewed Lab Protocol articles, which describe protocols hosted on protocols.io. Read more information on sharing protocols at https://plos.org/protocols?utm_medium=editorial-emailutm_source=authorlettersutm_campaign=protocols.

We look forward to receiving your revised manuscript.

Kind regards,

Hernâni Gerós, PhD

Academic Editor

PLOS ONE

Journal Requirements:

Reviewers' comments:

Reviewer's Responses to Questions

**Comments to the Author**

1. If the authors have adequately addressed your comments raised in a previous round of review and you feel that this manuscript is now acceptable for publication, you may indicate that here to bypass the “Comments to the Author” section, enter your conflict of interest statement in the “Confidential to Editor” section, and submit your "Accept" recommendation.

Reviewer #1: (No Response)

Reviewer #2: (No Response)

2. Is the manuscript technically sound, and do the data support the conclusions?

Reviewer #1: Partly

Reviewer #2: Yes

3. Has the statistical analysis been performed appropriately and rigorously? 

Reviewer #1: Yes

Reviewer #2: Yes

4. Have the authors made all data underlying the findings in their manuscript fully available?

Reviewer #1: Yes

Reviewer #2: Yes

5. Is the manuscript presented in an intelligible fashion and written in standard English?

Reviewer #1: Yes

Reviewer #2: Yes

6. Review Comments to the Author

Reviewer #1: Dear Authors,

Your revised PlosOne manuscript on Peronospora transporter gene expression was extended for my peer-review. Thank you for your work on this submission. At this stage, I regret to point out that not all comments were addressed satisfactorily or in uniform manner, and new issues arose. All those are pointed out below, with bylines identifying their occurrence. I am disappointed to see many inconsistencies throughout the manuscript, which implies inattention to detail, which in turn is not reflecting well on the Authors and quality of their work.

Answer to Reviewers (Disease vs. Pathogen)

Still present in this version of the manuscript, although Authors responded that it was corrected throughout. l.52,505,506.

"while" in the meaning of "whereas" - still present despite Authors response of correcting throughout.

RNA isolation / quality checks not described briefly at all - just refers to a previous paper. Mentioning a kit

name+manufacturer and whether their protocol was followed is preferred over referring to another (possibly paywalled) publication. That's against Open Science.

Authors agreed that "significant" in the meaning of "substantial" or similar was not a good word choice, yet still present in R1.

81 Further insights can add to context here: DOI 10.1016/bs.adgen.2020.03.001; 10.1101/2020.02.12.941195

116 leaves( )with

104 vs 119 unit inexistency

107 of( )-5

179 several (add: possible; conceivable; or similar)

188 the most (Consider: the highest number of)

192 Inconsistent use of serial comma (a.k.a., Oxford comma)

206 the top 50 (Here and in similar places; Correct throughout for the full meaning)

209;228;264 Most (Majority)

212,213 tubulin B; dehydrogenase alpha etc (Absent from Table 1. Or inconsistently named).

214 [MAJOR] The title should indicate what is shown in this table. What do all these numbers represent and how were they generated? Do not make the reader look for explanations, make this a reader-friendly manuscript.

221 The (expression rates of)

222 Inconsistent double spacing

235;411 significant (see Response to Reviewers comment)

240 5 for 5 the DPI (what is meant here?)

247 Some (What criteria were applied to select those "some"?)

251 An exemplary way to define and use throughout for the top X genes

252 timepoints (were "stages of infection" or similar meant here and in 254?)

254 [MAJOR] 7 is most definitely NOT the most of 16. Rephrase for precision.

262-267 [MAJOR] Is this passage referring to [47] or to the results of this very study? Rephrase for clarity.

264 91 pathways (data not shown or [47]?)

292 Abrahamian - inconsistent citation style

451-452 3-fold higher compared to (correct the grammar please; was " than the" meant here?)

480,505,506 "downy mildew" in the meaning of the causal pathogen(s)

496 they plays role (correct the grammar please)

504 pathogens . (spurious space)

References - incosistencies in journal names; pathogen italics, etc.

Reviewer #2: Reviewer: Dear editor and authors,

The authors developed in this study a comparison of gene expression of transporter proteins of three different Peronospora species during infection. The manuscript is simple, clear, well structured and written. For the study, they performed the infection experiments for one species and used two different public databases of published experiments. Therefore, this study is partially based on original scientific research but the analysis of the public databases is slightly interesting. The standard of the experiments is of high degree and the methods are sufficiently detailed, as well as the statistical analysis. The conclusions are based in the results and are well presented.

In the next line I present a series of recommendations and questions to debate with the authors.

• Line 26: This type of discussion is not adequate for the abstract. This specific line is a summary of our data in two different sentences. These data are mentioned in more detail in the discussion section.

RESPONSE: Line 29 “Several of the 16 highly-expressed transporter…will need to be confirmed in future experiments.” The sentences are these. This is a discussion sentence that do not add relevant information for the abstract, specially the second one which is innocuous for the discussion. Please remove the sentences. You can wrap it up just with the last sentence.

• What is the rational of study these 3 pathogen species? Downy mildew disease causes major losses on economically important plants, such as those in this study; it is important to find out more detail about how these pathogens infect their hosts. In addition, we thought it would be of interest to analyze transporter gene expression in these three pathosystems to see if there are similarities or differences in the most highly expressed transporter genes during host infection.

RESPONSE: So, the authors just selected downy mildew to study and then randomly selected 3 species affected by it.

• Can the authors explain the discrepancies regarding the days after inoculation of the samples? The studies of the basil, spinach and tobacco were independently completed. However, we felt that we could analyze this data together in this paper since the early timepoint (2 or 3 DPI) or the late timepoint (5, 6, or 7 DPI) only deviated one day or less than one day.

RESPONSE: I am not completely comfortable with this discrepancy, however is plausible that the physiological stage of the pathogen is not affected in such a manner that flaw this comparison.

• I cannot understand the value of the bars graphs presented in figure 1 without the names of the genes. We think it would be difficult to include the names of 50 genes on the horizontal axis. Readers would be able to find the gene name using the supplementary tables S3-S5.

RESPONSE: If you do not have the name of the genes what is the real purpose of the graphic? To have “one more result”? If is not informative it should not be included. At least use your nomenclature (the numbers, 1, 2, etc) to identify the genes in the respective supplementary table.

• The gene code presented in table 1 is the “same” for homologous genes? Is confusing the nomenclature in the text differs from the table. Maybe you can opt to introduce the specie gene name and its homologue. Admittedly it was difficult to identify a system for naming homologous genes. We settled on using the P. effusa RMX or RQM number as the homologous gene identifier in each Peronospora species. Readers can notice that the RMX/RQM naming system is used in Tables 2-4 for cross comparison. We think that we went through the Tables sequentially in the manuscript and introduce protein designations in the text (which were species specific) as needed. Moreover, Reviewer #1 did not mention a problem with our homologous gene identifiers.

RESPONSE: Is not a problem, is just confusing, is difficult to keep up.

• Line 215: “The TPM level of the MC gene PBEL_00715, as well as that of the actin gene, decreased from 3 to 6 DPI.” Why is this line relevant? Are these the only genes that decrease from 3 to 6 dpi? We thought it was interesting to find the expression (TPM level) of a transporter gene to decrease as the pathogen grew throughout the host. This is why we mentioned it.

RESPONSE: But it was the only gene that reduced the expression or the only transporter?

• Why did you choose Phytophthora infestans for comparison? We chose to compare our data with Phy. infestans because there was a paper measuring transporter gene expression during infection on tomato plants. We were curious to find any similarities in the expression patterns of the transporter genes. Phy. infestans is an oomycete but not an obligate plant pathogen.

RESPONSE: But Phytophthora infestans is a hemibiotrophic pathogen. Even just using the biotrophic phase is not completely suitable.

The authors just need to adjust a few things to complete the publication.

7. PLOS authors have the option to publish the peer review history of their article (what does this mean?). If published, this will include your full peer review and any attached files.

Reviewer #1: **Yes: **Marcin Nowicki

Reviewer #2: No

---

## [Author Response · Author response to Decision Letter 1]

3 Apr 2023

PONE-D-22-25502R1

A comparison of transporter gene expression in three species of Peronospora plant pathogens during host infection

PLOS ONE

Dear Dr. Johnson,

Thank you for submitting your manuscript to PLOS ONE. After careful consideration, we feel that it has merit but does not fully meet PLOS ONE’s publication criteria as it currently stands. Therefore, we invite you to submit a revised version of the manuscript that addresses the points raised during the review process.

All responses by the authors are highlighted in red. 

Please, improve the abstract, as reviewer #2 suggests, by removing of modifying the following two sentences "Several of the 16 highly-expressed transporter genes identified in this study were also highly-expressed in Phytophthora infestans during biotrophic infection of tomato leaves as measured in a previously published study. Comparisons to other studied transporters indicated that most of the 16 highly-expressed transporter genes have a putative function that will need to be confirmed in future experiments." These sentences were removed from the abstract. Also, do not forget to clarify figure 1 to make it informative, as suggested by the reviewer #2. We removed the bar graphs from figure 1 as recommended by reviewer #2. Please, also carefully address the remaining remarks of the reviewer #1. We addressed these remarks below.

Other changes in the R2 version not pointed out by the reviewers:

Numbering is from the "clean" manuscript

In the abstract

27 highly expressed, not highly-expressed

In the last paragraph of section titled "Differential expression of Peronospora transporter genes during interactions between oomycete pathogens and their susceptible hosts"

269 each plant host or may be due to variations in metabolism in each pathogen.

Comma after host removed

If applicable, we recommend that you deposit your laboratory protocols in protocols.io to enhance the reproducibility of your results. Protocols.io assigns your protocol its own identifier (DOI) so that it can be cited independently in the future. For instructions see: https://journals.plos.org/plosone/s/submission-guidelines#loc-laboratory-protocols. Additionally, PLOS ONE offers an option for publishing peer-reviewed Lab Protocol articles, which describe protocols hosted on protocols.io. Read more information on sharing protocols at https://plos.org/protocols?utm_medium=editorial-emailutm_source=authorlettersutm_campaign=protocols.

We look forward to receiving your revised manuscript.

Kind regards,

Hernâni Gerós, PhD

Academic Editor

PLOS ONE

Journal Requirements:

We checked all the cited literature and none of these papers were retracted to our knowledge. The paper by Constantinescu could not be found in an electronic form. A colleague (Dr. Joanne Crouch) provided this article for us. It has been cited 97 times according to Google Scholar (as of 3-13-23). In a recent paper (Validation of five Peronospora species names; Mycotaxon, Volume 136(4), October-December 2021, pp. 785-788) Davis and Crouch write this about the paper written by Constantinescu: "Constantinescu (1991) provides the most comprehensive list of Peronospora names and is an invaluable resource to those studying the genus."

Not much information regarding the legitimacy of Raabe et al. 1952 could be found. Therefore, a modern article focused on spinach downy mildew disease was used in its place (Kandel et al. 2019 Plant Disease 103: 791-803). 

Reviewers' comments:

Reviewer's Responses to Questions

Comments to the Author

1. If the authors have adequately addressed your comments raised in a previous round of review and you feel that this manuscript is now acceptable for publication, you may indicate that here to bypass the “Comments to the Author” section, enter your conflict of interest statement in the “Confidential to Editor” section, and submit your "Accept" recommendation.

Reviewer #1: (No Response)

Reviewer #2: (No Response)

2. Is the manuscript technically sound, and do the data support the conclusions?

Reviewer #1: Partly

Reviewer #2: Yes

3. Has the statistical analysis been performed appropriately and rigorously? 

Reviewer #1: Yes

Reviewer #2: Yes

4. Have the authors made all data underlying the findings in their manuscript fully available?

Reviewer #1: Yes

Reviewer #2: Yes

5. Is the manuscript presented in an intelligible fashion and written in standard English?

Reviewer #1: Yes

Reviewer #2: Yes

6. Review Comments to the Author

Reviewer #1: Dear Authors,

Your revised PlosOne manuscript on Peronospora transporter gene expression was extended for my peer-review. Thank you for your work on this submission. At this stage, I regret to point out that not all comments were addressed satisfactorily or in uniform manner, and new issues arose. All those are pointed out below, with bylines identifying their occurrence. I am disappointed to see many inconsistencies throughout the manuscript, which implies inattention to detail, which in turn is not reflecting well on the Authors and quality of their work.

Answer to Reviewers (Disease vs. Pathogen)

Still present in this version of the manuscript, although Authors responded that it was corrected throughout. l.52,505,506. All these instances were corrected. We also searched the entire document for occurrences of “downy mildew” to make sure that the usage was correct.

"while" in the meaning of "whereas" - still present despite Authors response of correcting throughout. We checked the entire manuscript and replaced “while” with “whereas” in all instances.

RNA isolation / quality checks not described briefly at all - just refers to a previous paper. Mentioning a kit name + manufacturer and whether their protocol was followed is preferred over referring to another (possibly paywalled) publication. That's against Open Science. A brief description of the RNA isolation procedure and subsequent quality checks were added to the manuscript.

Authors agreed that "significant" in the meaning of "substantial" or similar was not a good word choice, yet still present in R1. “Significant” has been replaced with “substantial” in all locations in the manuscript.

81 Further insights can add to context here: DOI 10.1016/bs.adgen.2020.03.001; 10.1101/2020.02.12.941195 We added two sentences that discussed some relative points about metabolic networks of obligate plant pathogens from the preprint manuscript by Rodenburg et al. 2020 (10.1101/2020.02.12.941195). The book chapter by McGowan and Fitzpatrick (2020, 10.1016/bs.adgen.2020.03.001) did not talk much about metabolic networks, so no text was added to the manuscript here.

116 leaves( )with. This typo was fixed.

104 vs 119 unit inexistency. The unit mL was changed to ml. 

107 (170) of( )-5. This typo was fixed.

179 several (add: possible; conceivable; or similar). “Several” was replaced with “similar”.

188 the most (Consider: the highest number of). The change was made as recommended.

192 Inconsistent use of serial comma (a.k.a., Oxford comma). A comma was added after “symporter”.

206 the top 50 (Here and in similar places; Correct throughout for the full meaning). More description was added here and in all locations, including Figure 2.

209;228;264 Most (Majority). “Most” was replaced with “The class of transporters that was most frequently found in the top 50 most highly expressed genes” for the P effusa data. A similar phrase was added to describe the most frequently found transporter class for the P belbahrii and P tabacina transporter gene analyses. Majority substituted for most at line 264.

212,213 tubulin B; dehydrogenase alpha etc (Absent from Table 1. Or inconsistently named). Corrections were made for these two gene names in the text as well as in Table 2. 

214 [MAJOR] The title should indicate what is shown in this table. What do all these numbers represent and how were they generated? Do not make the reader look for explanations, make this a reader-friendly manuscript. The title of Table 2 was modified. An explanation about the gene names was added to the Table 2 legend.

221 The (expression rates of). This sentence was changed as recommended.

222 Inconsistent double spacing. This double spacing was corrected.

235;411 significant (see Response to Reviewers comment). “Significant” was replaced with “substantial” at these two locations.

240 5 for 5 the DPI (what is meant here?). The order of the words was changed here.

247 Some (What criteria were applied to select those "some"?). The criteria were described in the modified sentence.

251 An exemplary way to define and use throughout for the top X genes. Noted.

252 timepoints (were "stages of infection" or similar meant here and in 254?). Timepoints was rephrased in both instances.

254 [MAJOR] 7 is most definitely NOT the most of 16. Rephrase for precision. The following edit was done: “in common, seven were MCs, as shown in”.

262-267 [MAJOR] Is this passage referring to [47] or to the results of this very study? Rephrase for clarity. The Rodenberg et al. paper [47] was cited several times as needed in this revised paragraph. 

264 91 pathways (data not shown or [47]?). The 91 pathways were from [47]. The sentence was modified accordingly.

292 Abrahamian - inconsistent citation style. The data of Abrahamian et al. was not cited at this location because Reviewer #2 suggested to remove all the Phytophthora infestans data from Table 3.

451-452 3-fold higher compared to (correct the grammar please; was " than the" meant here?). The grammar was corrected here, and some additional text was added for clarity.

480,505,506 "downy mildew" in the meaning of the causal pathogen(s). For each occurrence, “pathogen” was added after “downy mildew”.

496 they plays role (correct the grammar please). In this sentence, “they” was replaced with “each transporter”. 

504 pathogens . (spurious space). This extra space was removed.

References - inconsistencies in journal names; pathogen italics, etc. The references were formatted for the standard PLoS One style using EndNote software.

Reviewer #2: Reviewer: Dear editor and authors,

The authors developed in this study a comparison of gene expression of transporter proteins of three different Peronospora species during infection. The manuscript is simple, clear, well structured and written. For the study, they performed the infection experiments for one species and used two different public databases of published experiments. Therefore, this study is partially based on original scientific research but the analysis of the public databases is slightly interesting. The standard of the experiments is of high degree and the methods are sufficiently detailed, as well as the statistical analysis. The conclusions are based in the results and are well presented.

In the next line I present a series of recommendations and questions to debate with the authors.

• Line 26: This type of discussion is not adequate for the abstract. This specific line is a summary of our data in two different sentences. These data are mentioned in more detail in the discussion section.

RESPONSE: Line 29 “Several of the 16 highly-expressed transporter…will need to be confirmed in future experiments.” The sentences are these. This is a discussion sentence that do not add relevant information for the abstract, specially the second one which is innocuous for the discussion. Please remove the sentences. You can wrap it up just with the last sentence. We removed the two sentences from the abstract.

• What is the rational of study these 3 pathogen species? Downy mildew disease causes major losses on economically important plants, such as those in this study; it is important to find out more detail about how these pathogens infect their hosts. In addition, we thought it would be of interest to analyze transporter gene expression in these three pathosystems to see if there are similarities or differences in the most highly expressed transporter genes during host infection.

RESPONSE: So, the authors just selected downy mildew to study and then randomly selected 3 species affected by it. Our government established research project was focused on downy mildew disease at the time. That is why we chose to study downy mildew. The three species were chosen because transcriptome data of two species were available for analysis. The corresponding author published research on the basil downy mildew pathogen transcriptome in 2022. Transcriptome data for the spinach downy mildew pathogen was in NCBI GenBank. The current study generated transcriptome data for the tobacco downy mildew pathogen.

• Can the authors explain the discrepancies regarding the days after inoculation of the samples? The studies of the basil, spinach and tobacco were independently completed. However, we felt that we could analyze this data together in this paper since the early timepoint (2 or 3 DPI) or the late timepoint (5, 6, or 7 DPI) only deviated one day or less than one day.

RESPONSE: I am not completely comfortable with this discrepancy, however is plausible that the physiological stage of the pathogen is not affected in such a manner that flaw this comparison. Noted.

• I cannot understand the value of the bars graphs presented in figure 1 without the names of the genes. We think it would be difficult to include the names of 50 genes on the horizontal axis. Readers would be able to find the gene name using the supplementary tables S3-S5.

RESPONSE: If you do not have the name of the genes what is the real purpose of the graphic? To have “one more result”? If is not informative it should not be included. At least use your nomenclature (the numbers, 1, 2, etc) to identify the genes in the respective supplementary table. The bar graphs were removed from Figure 1, and the figure was reformatted.

• The gene code presented in table 1 is the “same” for homologous genes? Is confusing the nomenclature in the text differs from the table. Maybe you can opt to introduce the specie gene name and its homologue. Admittedly it was difficult to identify a system for naming homologous genes. We settled on using the P. effusa RMX or RQM number as the homologous gene identifier in each Peronospora species. Readers can notice that the RMX/RQM naming system is used in Tables 2-4 for cross comparison. We think that we went through the Tables sequentially in the manuscript and introduce protein designations in the text (which were species specific) as needed. Moreover, Reviewer #1 did not mention a problem with our homologous gene identifiers.

RESPONSE: Is not a problem, is just confusing, is difficult to keep up. Noted. We also edited some of the manuscript text where it referred to RMX or RQM genes for clarity. RMX or RQM gene names are used throughout the manuscript until the section “In depth transporter protein analysis”. 

• Line 215: “The TPM level of the MC gene PBEL_00715, as well as that of the actin gene, decreased from 3 to 6 DPI.” Why is this line relevant? Are these the only genes that decrease from 3 to 6 dpi? We thought it was interesting to find the expression (TPM level) of a transporter gene to decrease as the pathogen grew throughout the host. This is why we mentioned it.

RESPONSE: But it was the only gene that reduced the expression or the only transporter? This sentence was deleted, as it is not that necessary for the whole story.

• Why did you choose Phytophthora infestans for comparison? We chose to compare our data with Phy. infestans because there was a paper measuring transporter gene expression during infection on tomato plants. We were curious to find any similarities in the expression patterns of the transporter genes. Phy. infestans is an oomycete but not an obligate plant pathogen.

RESPONSE: But Phytophthora infestans is a hemibiotrophic pathogen. Even just using the biotrophic phase is not completely suitable. The entire column with P. infestans data, as well as the explanatory text, was removed from the table and manuscript, respectively.

The authors just need to adjust a few things to complete the publication. Thank you for your assistance in making our manuscript better.

7. PLOS authors have the option to publish the peer review history of their article (what does this mean?). If published, this will include your full peer review and any attached files.

Do you want your identity to be public for this peer review? For information about this choice, including consent withdrawal, please see our Privacy Policy.

Reviewer #1: Yes: Marcin Nowicki

Reviewer #2: No

---

## [Decision Letter · Decision Letter 2]

28 Apr 2023

A comparison of transporter gene expression in three species of Peronospora plant pathogens during host infection

PONE-D-22-25502R2

Dear Dr. Johnson,

We’re pleased to inform you that your manuscript has been judged scientifically suitable for publication and will be formally accepted for publication once it meets all outstanding technical requirements.

Kind regards,

Hernâni Gerós, PhD

Academic Editor

PLOS ONE

Additional Editor Comments (optional):

Reviewers' comments:

Reviewer's Responses to Questions

**Comments to the Author**

1. If the authors have adequately addressed your comments raised in a previous round of review and you feel that this manuscript is now acceptable for publication, you may indicate that here to bypass the “Comments to the Author” section, enter your conflict of interest statement in the “Confidential to Editor” section, and submit your "Accept" recommendation.

Reviewer #1: All comments have been addressed

Reviewer #2: All comments have been addressed

2. Is the manuscript technically sound, and do the data support the conclusions?

Reviewer #1: Yes

Reviewer #2: Yes

3. Has the statistical analysis been performed appropriately and rigorously? 

Reviewer #1: Yes

Reviewer #2: Yes

4. Have the authors made all data underlying the findings in their manuscript fully available?

Reviewer #1: Yes

Reviewer #2: Yes

5. Is the manuscript presented in an intelligible fashion and written in standard English?

Reviewer #1: Yes

Reviewer #2: Yes

6. Review Comments to the Author

Reviewer #1: (No Response)

Reviewer #2: The authors have addressed the questions that were raised by reviewer #2, therefore the manuscript is in a acceptable form to publish.

7. PLOS authors have the option to publish the peer review history of their article (what does this mean?). If published, this will include your full peer review and any attached files.

Reviewer #1: **Yes: **Marcin Nowicki

Reviewer #2: No

---

## [Editor Report · Acceptance letter]

23 May 2023

PONE-D-22-25502R2 

A comparison of transporter gene expression in three species of *Peronospora* plant pathogens during host infection 

Dear Dr. Johnson:

I'm pleased to inform you that your manuscript has been deemed suitable for publication in PLOS ONE. Congratulations! Your manuscript is now with our production department. 

Kind regards, 

on behalf of

Dr. Hernâni Gerós 

Academic Editor

PLOS ONE